# Variance-Reduced $(\varepsilon, \delta)-$Unlearning Using Forget Set Gradients

**Martin Van Waerebeke** [1]  **Giovanni Neglia** [2]  **Kevin Scaman** [1]  **Marco Lorenzi** [2]  **El-Mahdi El-Mhamdi** [3]

## Abstract

In machine unlearning, $(\varepsilon, \delta)-$unlearning is a popular framework that provides formal guarantees on the effectiveness of the removal of a subset of training data, the *forget set*, from a trained model. For strongly convex objectives, existing first-order methods achieve $(\varepsilon, \delta)-$unlearning, but only use the forget set to calibrate injected noise, never as a direct optimization signal. In contrast, efficient empirical heuristics often exploit the forget samples (e.g., via gradient ascent) but come with no formal unlearning guarantees. We bridge this gap by presenting the Variance-Reduced Unlearning (*VRU*) algorithm. To the best of our knowledge, *VRU* is the first first-order algorithm that directly includes forget set gradients in its update rule, while provably satisfying $(\varepsilon, \delta)-$unlearning. We establish the convergence of *VRU* and show that incorporating the forget set yields strictly improved rates, *i.e.,* a better dependence on the achieved error compared to existing first-order $(\varepsilon, \delta)-$unlearning methods. Moreover, we prove that, in a low-error regime, *VRU* asymptotically outperforms any first-order method that ignores the forget set. Experiments corroborate our theory, showing consistent gains over state-of-the-art certified unlearning methods and empirical baselines that explicitly leverage the forget set.

## 1. Introduction

Machine Unlearning (MU) aims at efficiently removing the influence of a subset of training examples (the *forget set*) from a trained model so that, after unlearning, the model is equivalent to one obtained by retraining from scratch on the remaining data (the *retain set*). The objective of MU is thus threefold: i) ensuring erasure of the forget set, ii) preserving model utility on the retain set, and iii) achieving substantial computational savings with respect to retraining from scratch.

To formalize the notion of "erasure," the community has developed the framework of $(\varepsilon, \delta)-$unlearning (Ginart et al., 2019), based on the framework of differential privacy (Dwork & Roth, 2014). More precisely, $(\varepsilon, \delta)-$unlearning provides statistical indistinguishability between the distributions of models obtained with a given unlearning procedure, and the one obtained by retraining on the retain set. Within this setting, first order methods based on variants of gradient descent are popular unlearning approaches, as their computational complexity scales well with the model size (Allouah et al., 2024a; Sepahvand et al., 2025; Koloskova et al., 2025). These approaches rely primarily on fine-tuning on the retain set, and on suitable noise injection to guarantee the erasure of the forget set (Neel et al., 2021).

Nevertheless, current theories for $(\varepsilon, \delta)-$unlearning present relevant *theoretical* and *practical* limitations, as they make limited use of the forget set during the unlearning procedure, using it primarily to calibrate the injected noise rather than as an optimization signal.

From the theoretical perspective, it has recently been shown that the *gradient on the forget set* drives the steepest descent direction for the unlearning loss (Huang et al., 2024). Moreover, first order methods that do not depend on the forget set are less efficient than retraining from scratch in the low error regime (Van Waerebeke et al., 2025). From the practical standpoint, several MU methods, not backed by $(\varepsilon, \delta)$ guarantees, exploit the forget set for the unlearning optimization strategy, for example by performing gradient ascent steps on the forget set, or by fine-tuning on random labels (Fan et al., 2023). While these methods are empirical and lack formal guarantees, they have been shown to outperform certified $(\varepsilon, \delta)-$unlearning methods in popular benchmarks (Maini et al., 2024; Shi et al., 2024; Li et al., 2024).

**Contributions.** In this work we reconcile first order $(\varepsilon, \delta)-$unlearning algorithms with the effective use of the forget set during the unlearning procedure. To this end, we extend the family of first order $(\varepsilon, \delta)-$unlearning approaches with a novel method, *V*ariance-*R*educed *U*nlearning (*VRU*), and demonstrate its superiority over the state-of-the-art on both theoretical and practical aspects.

[1]INRIA Paris [2]INRIA Sophia Antipolis [3]CMAP - École polytechnique. Correspondence to: Martin Van Waerebeke <martin.van-waerebeke@inria.fr>.

*Proceedings of the $43^{rd}$ International Conference on Machine Learning*, Seoul, South Korea. PMLR 306, 2026. Copyright 2026 by the author(s).

In particular, we:

- Propose *VRU*, the first first-order $(\varepsilon, \delta)-$unlearning method that includes gradient ascent on the forget set to improve unlearning efficiency.

- Analyze *VRU* for strongly-convex losses, establishing convergence rates that improve over existing first-order $(\varepsilon, \delta)-$unlearning approaches.

- Prove that *VRU* asymptotically beats any first-order $(\varepsilon, \delta)-$unlearning algorithm that does not use the forget-set.

- Practically validate the effectiveness of *VRU* against both $(\varepsilon, \delta)-$unlearning and empirical methods.

## 2. Related Work

The domain of machine unlearning (MU) studies how to efficiently remove the influence of a designated subset of training examples from a learned model, with the goal of matching the model that would be obtained by retraining from scratch on the remaining data, at a substantially lower computational cost. Much progress has been made recently, both towards new unlearning methods (Guo et al., 2024; Schoepf et al., 2025) and new ways to evaluate them (Maini et al., 2024; Hayes et al., 2025; Lu et al., 2025).

*Unlearning methods are commonly categorized as either exact or approximate.* The former return a model identical to the one obtained by retraining from scratch on the retain set, whereas the latter aim to closely approximate the retrained solution. Exact methods offer ideal privacy guarantees, but usually require a modification of the training process, often through tree-based approaches (Ullah et al., 2021; Ullah & Arora, 2023) or sharding (Bourtoule et al., 2021; Yan et al., 2022; Wang et al., 2023), limiting their scope.

*Approximate MU algorithms can be divided into empirical and certified methods.* Empirical approaches prioritize efficiency and broad applicability, but do not provide explicit unlearning guarantees, whereas certified methods provably satisfy a formal definition of unlearning. Several notions of certified unlearning have been proposed. Early work focused on minimizing the KL-divergence with respect to a retrained model (Golatkar et al., 2020b;a; 2021; Jin et al., 2023). More recent definitions consider alternatives such as cosine similarity with retraining (Melamed et al., 2025) or more sample-specific criteria (Sepahvand et al., 2025). However, most certified unlearning methods are based on the notion of $(\varepsilon, \delta)-$unlearning (Ginart et al., 2019), with the aim of achieving statistical indistinguishability between the post-unlearning model and a model retrained from scratch. To meet this guarantee, existing algorithms rely on either first-order (Neel et al., 2021) or second-order (Guo et al.,

2020) updates. Second-order methods exploit information about the local curvature of the loss function, while first-order methods typically scale better to high-dimensional models.

*First-order MU methods that achieve $(\varepsilon, \delta)$-unlearning rely on a combination of noise injection and gradient-based fine-tuning on the retain set.* Recent work has established convergence and utility guarantees for such methods under various assumptions (Neel et al., 2021; Mu & Klabjan, 2024; Chien et al., 2024; Van Waerebeke et al., 2025; Koloskova et al., 2025; Benarroch et al., 2026). These approaches differ primarily on how they combine the two core operations: noise injection and fine-tuning on the retain set. Existing strategies include (i) fine-tuning on the retain set followed by output perturbation (Neel et al., 2021; Allouah et al., 2024b), (ii) adding noise to the model and then fine-tuning (Fraboni et al., 2024; Van Waerebeke et al., 2025; Mu & Klabjan, 2024), and (iii) injecting noise at each gradient step during fine-tuning (Chien et al., 2024; Sepahvand et al., 2025; Benarroch et al., 2026). These works also differ in the type of guarantee they provide. Our theoretical comparisons focus on methods that yield a full utility–privacy–complexity characterization of the unlearned model in the strongly convex setting. By contrast, noisy fine-tuning methods such as those of Chien et al. (2024) study related certified unlearning procedures under broader assumptions, but do not typically provide the same excess-risk characterization after a given number of unlearning steps.

*Other subdomains of the literature rely on the forget set to achieve approximate unlearning.* In particular, most empirical methods and most second-order certified approaches exploit the forget set to their advantage. Empirical methods often apply gradient ascent on the forget set (Chen & Yang, 2023; Yao et al., 2024b; Pang et al., 2025), but they also leverage it in other ways: discouraging the model from reproducing the original predictions on the forget set (Kurmanji et al., 2024), training on fake or homogeneous labels (Fan et al., 2023), replacing sensitive forget-set information with generic one (Zhang et al., 2024) or adversarial (Yao et al., 2024a) content. Second-order methods typically combine the forget-set gradient with the inverse Hessian to form a Newton-like unlearning update (Guo et al., 2020; Sekhari et al., 2021; Golatkar et al., 2020a). Recent work accelerates these updates by avoiding explicit inverse-Hessian computation in an online-learning setting (Qiao et al., 2024).

*Forget-set ascent steps are crucial for designing efficient MU procedures.* Recent works have provided theoretical evidence that first-order MU algorithms must incorporate forget-set–based updates to achieve efficient unlearning. Under a local second-order approximation, Huang et al. (2024) decomposes the optimal unlearning update, proving that it is a weighted sum of a term involving the forget set gradi-

ent, and the usual retain set gradient. This is corroborated by Ding et al. (2024), who showcases the limits of only minimizing the loss on $\mathcal{D}_r$, and suggests the addition of a term pulling the model away from the forget-set optimum. Moreover, Van Waerebeke et al. (2025) analyses the algorithmic complexity of MU methods compared to re-training and prove that, when optimizing to sufficiently small errors, any first-order unlearning algorithm that does not rely on forget-set gradients cannot outperform retraining from scratch asymptotically.

## 3. Problem Statement

We first introduce the notation used in our analysis.

We consider a supervised learning setting with a model parameter $\boldsymbol{\theta} \in \mathbb{R}^d$ and a loss $\ell : \mathbb{R}^d \times \mathbb{R}^s \to \mathbb{R} \cup \{+\infty\}$. Let $\mathcal{D}_f$ and $\mathcal{D}_r$ denote the forget and retain data distributions over $\mathbb{R}^s$, respectively. The original training distribution is the mixture $\mathcal{D} := r_f \mathcal{D}_f + (1 - r_f) \mathcal{D}_r$, where $r_f \in (0, 1)$ represents the fraction of data to be unlearned. For any distribution $\mathcal{D}$, we define the population risk as

$$\mathcal{L}(\boldsymbol{\theta}, \mathcal{D}) := \mathbb{E}_{\xi \sim \mathcal{D}}[\ell(\boldsymbol{\theta}, \xi)]. \tag{1}$$

We recall that a differentiable function $f : \mathbb{R}^d \to \mathbb{R}$ is $\mu$-*strongly convex* if $f(\boldsymbol{\theta}') \geq f(\boldsymbol{\theta}) + \langle \nabla f(\boldsymbol{\theta}), \boldsymbol{\theta}' - \boldsymbol{\theta} \rangle + \frac{\mu}{2}\|\boldsymbol{\theta}' - \boldsymbol{\theta}\|^2$; $\beta$-*smooth* if $\|\nabla f(\boldsymbol{\theta}) - \nabla f(\boldsymbol{\theta}')\| \leq \beta\|\boldsymbol{\theta} - \boldsymbol{\theta}'\|$; and $L$-*Lipschitz* if $|f(\boldsymbol{\theta}) - f(\boldsymbol{\theta}')| \leq L\|\boldsymbol{\theta} - \boldsymbol{\theta}'\|$, for all $\boldsymbol{\theta}, \boldsymbol{\theta}'$. We impose these regularity conditions on the sample loss:

**Assumption 3.1** (Loss regularity). For any data point $\xi \in \mathbb{R}^s$, the function $\ell(\cdot, \xi)$ is $\mu$-strongly convex, $\beta$-smooth, and $L$-Lipschitz with respect to $\boldsymbol{\theta}$.

We write $\mathcal{F}$ for the class of functions satisfying Ass. 3.1. These assumptions are standard in the machine unlearning literature, particularly when analyzing finite-time convergence alongside privacy guarantees (Chourasia & Shah, 2023; Huang & Canonne, 2023; Allouah et al., 2024b; Van Waerebeke et al., 2025). Under Ass. 3.1, let $\boldsymbol{\theta}^*$ (resp. $\boldsymbol{\theta}_r^*$) denote the unique minimizer of $\mathcal{L}(\cdot, \mathcal{D})$ (resp. $\mathcal{L}(\cdot, \mathcal{D}_r)$), and let $\kappa_\ell := \beta/\mu$ denote the condition number. Finally, note that $\mu$-strong convexity and $L$-Lipschitz continuity jointly imply that the loss has a bounded effective domain, with diameter at most $4L/\mu$.

A consequence of this bounded domain is a bound on the excess loss at initialization: for any model $\boldsymbol{\theta} \in \mathbb{R}^d$ that admits a finite loss, $\mathcal{L}(\boldsymbol{\theta}, \mathcal{D}) - \mathcal{L}(\boldsymbol{\theta}^*, \mathcal{D}) \leq \frac{L^2}{\mu} =: e_0$.

In some of our results, we compare *VRU* against *arbitrary* retraining/unlearning algorithms. This level of generality comes at the cost of additional, though still broad, assumptions on the distributions $\mathcal{D}_r$ and $\mathcal{D}_f$.

**Assumption 3.2** (Distributional assumptions). The retain distribution $\mathcal{D}_r$ is such that, for any $p \in [0, 1]$, there exists

a measurable $A \subset \mathbb{R}^s$ with $\mathbb{P}_{\xi \sim \mathcal{D}_r}[\xi \in A] = p$. Moreover, there exist disjoint measurable sets $S_r, S_f \subseteq \mathbb{R}^s$ with $\mathbb{P}_{\xi \sim \mathcal{D}_r}[\xi \in S_r] = 1$ and $\mathbb{P}_{\xi \sim \mathcal{D}_f}[\xi \in S_f] = 1$, and $(S_r \cup S_f)^c \neq \emptyset$.

Note that Assumption 3.2 holds, for instance, when $\mathcal{D}_r$ is absolutely continuous and the supports of $\mathcal{D}_r$ and $\mathcal{D}_f$ are disjoint and do not cover all of $\mathbb{R}^s$. This support separation arises naturally when retain and forget data are drawn from distinct subpopulations, as in class unlearning or toxic content removal.

**Class of Unlearning Algorithms.** We formally define the class of unlearning algorithms considered in this work.

**Definition 3.3** (Unlearning Algorithm). An unlearning algorithm $\mathcal{U} : \mathbb{N} \times \mathcal{F} \times \mathcal{P}(\mathbb{R}^s) \times \mathcal{P}(\mathbb{R}^s) \to \mathbb{R}^d$ is a (possibly randomized) procedure that takes as input (i) a number of iterations $T$, (ii) a loss function $\ell$, and access to samples from the (iii) retain distribution $\mathcal{D}_r$, and (iv) forget distribution $\mathcal{D}_f$. The output of the unlearning algorithm is a model $\boldsymbol{\theta}_T = \mathcal{U}(T, \ell, \mathcal{D}_r, \mathcal{D}_f) \in \mathbb{R}^d$. The algorithm is initialized at the optimum $\boldsymbol{\theta}^*$ which we omit from the notation. Initializing at $\boldsymbol{\theta}^*$, the minimizer of $\mathcal{L}(\cdot, \mathcal{D})$, is standard in the $(\varepsilon, \delta)$-unlearning literature (Guo et al., 2020; Yi & Wei, 2024; Chien et al., 2024; Van Waerebeke et al., 2025). It also reflects the practical unlearning setting, where the update procedure starts from the model trained on the full dataset. We denote by $\mathbb{U}$ the class of unlearning algorithms.

In contrast, a retraining algorithm $\mathcal{A} : \mathbb{N} \times \mathcal{P}(\mathbb{R}^s) \to \mathbb{R}^d$ is a (possibly randomized) training procedure that takes as input the retraining time $T$, the retain distribution $\mathcal{D}_r$, and outputs a model $\boldsymbol{\theta}_T = \mathcal{A}(T, \mathcal{D}_r) \in \mathbb{R}^d$ trained exclusively on the retain data. These algorithms represent training "from scratch", their initialization is thus random and will not be made explicit. Similarly, the number of iterations $T$ will be omitted when not of interest, for notation simplicity. We denote by $\mathbb{A}$ the class of all learning algorithms.

We adopt the standard definition of $(\varepsilon, \delta)$-unlearning, first proposed by Ginart et al. (2019).

**Definition 3.4** ($(\varepsilon, \delta)$-unlearning). An unlearning algorithm $\mathcal{U} \in \mathbb{U}$ satisfies $(\varepsilon, \delta)$-unlearning if there exists a retraining algorithm $\mathcal{A}$ such that, for any distributions $(\mathcal{D}_r, \mathcal{D}_f)$, and any measurable subset $S \subseteq \mathbb{R}^d$,

$$\mathbb{P}[\mathcal{U}(T, \ell, \mathcal{D}_r, \mathcal{D}_f) \in S] \leq e^\varepsilon \cdot \mathbb{P}[\mathcal{A}(\mathcal{D}_r) \in S] + \delta,$$
$$\mathbb{P}[\mathcal{A}(\mathcal{D}_r) \in S] \leq e^\varepsilon \cdot \mathbb{P}[\mathcal{U}(T, \ell, \mathcal{D}_r, \mathcal{D}_f) \in S] + \delta.$$

In other words, an unlearning method achieves $(\varepsilon, \delta)$-unlearning if the distribution of models that unlearned $\mathcal{D}_f$ is close to the distribution of models retrained from scratch on $\mathcal{D}_r$; the latter are perfectly private with respect to $\mathcal{D}_f$ since they have never been exposed to it.

We denote by $\mathbb{U}_{\varepsilon,\delta} \subseteq \mathbb{U}$ the class of all unlearning algorithms satisfying $(\varepsilon, \delta)$-unlearning, and by $\mathbb{U}_{\varepsilon,\delta}^r \subseteq \mathbb{U}_{\varepsilon,\delta}$ the subclass that does not access the forget set during the unlearning procedure. We refer to the latter as forget-set-free methods. We further define the privacy budget $\kappa_{\epsilon,\delta} := \varepsilon^{-1}\sqrt{2\log(2.5/\delta)}$ (Dwork & Roth, 2014), which relates the scale of the noise to the privacy parameters.

**Convergence times.** To evaluate the utility of unlearning algorithms, we adopt the standard notion of convergence time. This notion allows to link the largest accepted error of an unlearning procedure with the time it takes to achieve it (Allouah et al., 2024b; Van Waerebeke et al., 2025; Zou et al., 2025). For a given target error $e > 0$ and an unlearning algorithm $\mathcal{U} \in \mathbb{U}$, we define

$$T_e(\ell, \mathcal{U}) := \min_{T \in \mathbb{N}}\{T; \ \mathbb{E}[\mathcal{L}_r(\mathcal{U}(T, \ell, \mathcal{D}_r, \mathcal{D}_f)) - \mathcal{L}_r(\boldsymbol{\theta}_r^*)] \le e\},$$

where $\mathcal{L}_r(\cdot) := \mathcal{L}(\cdot, \mathcal{D}_r)$. Similarly, for a given target error $e > 0$ and a retraining algorithm $\mathcal{A} \in \mathbb{A}$, we define

$$T_e(\ell, \mathcal{A}) := \min_{T \in \mathbb{N}}\{T; \ \mathbb{E}[\mathcal{L}_r(\mathcal{A}(T, \mathcal{D}_r) - \mathcal{L}_r(\boldsymbol{\theta}_r^*)] \le e\}.$$

For our subsequent analysis, we define the error parameter

$$\nu_T := \sqrt{2h(T, \delta)}L\frac{1 + \kappa_\ell}{\mu\sqrt{T}}, \qquad (2)$$

where $h(T, \delta) := 1 + 624\left(\log(\log(T)) + \log(2/\delta)\right)$. This parameter is used to bound, with high probability, the distance between a *VRU* iterate and the optimum $\boldsymbol{\theta}_r^*$.

**Notation.** We denote by $B(\boldsymbol{\theta}, R)$ the closed Euclidean ball of radius $R$ around $\boldsymbol{\theta}$, and by $\mathrm{proj}_C(\boldsymbol{x}) = \mathrm{argmin}_{\boldsymbol{y} \in C} \|\boldsymbol{x} - \boldsymbol{y}\|_2$ the Euclidean projection of $\boldsymbol{x}$ onto a closed convex set $C$.

## 4. Main Results

In this section, we first introduce our Variance-Reduced Unlearning (*VRU*) algorithm and explain the intuition behind its gradient estimator (Section 4.1). We then establish its convergence rate (Section 4.2) and compare it to the state-of-the-art, proving significant speedup w.r.t. unlearning and retraining methods (Section 4.3). We finally provide a formal separation: *VRU* provably outperforms *any* $(\varepsilon, \delta)$-unlearning method that does not access the forget set (Section 4.4). All proofs are deferred to Appendix A.

### 4.1. Introducing the *VRU* Algorithm

We define the variance-reduced unlearning (*VRU*) algorithm, and its associated stochastic gradient estimator,

$$\widetilde{\nabla}(\boldsymbol{\theta}, \xi_r, \xi_f) = \nabla\ell(\boldsymbol{\theta}, \xi^r) \underbrace{- \nabla\ell(\boldsymbol{\theta}^*, \xi^r) - \frac{r_f}{1 - r_f}\nabla\ell(\boldsymbol{\theta}^*, \xi^f)}_{\text{correction term}},$$

$$(3)$$

where $\xi^f \sim \mathcal{D}_f$ and $\xi^r \sim \mathcal{D}_r$ are i.i.d. samples respectively drawn from the forget and retain distributions. The intuition behind this gradient estimator is relatively straightforward: to the usual stochastic gradient, we add a correction term with null expectation that significantly reduces the variance.

**Structure of the *VRU* algorithm.** Our proposed algorithm operates in two phases. First, we apply the Projected Stochastic Gradient Descent algorithm (PSGD, see *e.g.,* Alg. 9.11 in Garrigos & Gower (2023)) to our gradient estimator $\widetilde{\nabla}$ with decreasing step size $1/(\mu t)$. The projection is done onto the ball $B(\boldsymbol{\theta}^*, \frac{r_f}{1 - r_f}\frac{L}{\mu})$, which is guaranteed to contain the global optimum $\boldsymbol{\theta}_r^*$ (see Lemma A.1). Thus, the projection never moves the iterates away from the optimum. Finally, we apply noise to the optimization output to ensure $(\varepsilon, \delta)$-unlearning. Rapid convergence to the global minimizer $\boldsymbol{\theta}_r^\star$ implies that only a small amount of noise is required: the latter scales with $r_f$ and decreases as $1/\sqrt{T}$ with the number of optimization steps $T$. We provide a formal analysis in Appendix A.

**Unbiasedness of $\widetilde{\nabla}(\boldsymbol{\theta})$.** As $\boldsymbol{\theta}^*$ is the minimizer of the loss on $\mathcal{D}$, the gradients on the retain and forget sets cancel each other out at this specific point,

$$(1 - r_f)\mathbb{E}[\nabla\ell(\boldsymbol{\theta}^*, \xi^r)] = (1 - r_f)\nabla\mathcal{L}(\boldsymbol{\theta}^*, \mathcal{D}_r) \qquad (4)$$

$$= -r_f\nabla\mathcal{L}(\boldsymbol{\theta}^*, \mathcal{D}_f) = -r_f\mathbb{E}[\nabla\ell(\boldsymbol{\theta}^*, \xi^f)].$$

$$(5)$$

This ensures that our correction term has zero expectation, keeping the gradient estimator unbiased.

**Reduced variance of $\widetilde{\nabla}(\boldsymbol{\theta})$.** A natural approach to approximate $\boldsymbol{\theta}_r^*$ starting from $\boldsymbol{\theta}^*$ is to apply stochastic gradient descent on $\mathcal{D}_r$ using $\nabla\ell(\boldsymbol{\theta}, \xi_r)$. While the expected norm of this gradient is small near $\boldsymbol{\theta}^*$, at most $\frac{r_f}{1 - r_f}L$ (Eq. 5), it may still exhibit significant variance even when $r_f$ is small. This leads to slow convergence when fine-tuning on $\mathcal{D}_r$ alone, a common approach in certified unlearning methods.

The challenge lies in reducing this variance without having to compute full-batch gradients. To this end, we draw on the main idea behind the *SVRG* algorithm (Johnson & Zhang, 2013), which replaces the stochastic gradient with the difference $\nabla\ell(\boldsymbol{\theta}, \xi_r) - \nabla\ell(\boldsymbol{\theta}^*, \xi_r)$. Since the loss is smooth, this difference has a low variance when $\boldsymbol{\theta}$ and $\boldsymbol{\theta}^*$ are close. However, this introduces a bias, which *SVRG* corrects by adding a full-batch gradient at a certain anchor point, $\nabla\mathcal{L}_r(\boldsymbol{\theta}_{\text{anchor}})$, a computationally expensive operation that must be periodically recomputed as the iterates move away from the anchor, *i.e.,* $\boldsymbol{\theta}$ is far from $\boldsymbol{\theta}_{\text{anchor}}$.

Our key observation is that we can apply a similar approach in unlearning, and instead add the stochastic term $-\frac{r_f}{1 - r_f}\nabla\ell(\boldsymbol{\theta}^*, \xi_f)$, which has the same expectation as the

**Algorithm 1 VRU** (Variance Reduced Unlearning)

---

**Require:** Number of iterations $T$, trained model $\boldsymbol{\theta}^*$, loss function $\ell$, forget ratio $r_f \in (0,1)$, retain set $\mathcal{D}_r$, forget set $\mathcal{D}_f$, privacy budget $\kappa_{\epsilon,\delta}$

1: Set $\theta_1 = \theta^*$, compute $\nu_T$ (Eq. 2) and $R := \frac{r_f}{1-r_f}\frac{L}{\mu}$
2: **for** $t = 1$ to $T$ **do**
3:     Sample data points: $\xi_t^r \sim \mathcal{D}_r, \xi_t^f \sim \mathcal{D}_f$
4:     Compute variance-reduced gradient:

$$\tilde{\nabla}_t \leftarrow \nabla\ell(\boldsymbol{\theta}_t, \xi_t^r) - \nabla\ell(\boldsymbol{\theta}^*, \xi_t^r) - \frac{r_f}{1-r_f}\nabla\ell(\boldsymbol{\theta}^*, \xi_t^f)$$

5:     Update parameters and perform projection on the ball: $\boldsymbol{\theta}_{t+1} \leftarrow \text{proj}_{B(\boldsymbol{\theta}^*, R)}\left(\boldsymbol{\theta}_t - 1/\mu t\, \tilde{\nabla}_t\right)$
6: **end for**
7: Sample $Z \sim \mathcal{N}(0, I_d)$
8: Noise the model to ensure unlearning:

$$\tilde{\boldsymbol{\theta}}_T = \boldsymbol{\theta}_T + \left(\frac{r_f}{1-r_f}\right)\nu_T \kappa_{\epsilon,\delta} Z$$

9: **return** Final model $\tilde{\boldsymbol{\theta}}_T$

---

full gradient (Eq. 5). As $r_f$ represents a small portion of data, this term has much lower variance than $\nabla\ell(\boldsymbol{\theta}^*, \xi_r)$, allowing it to correct the bias without compromising the variance reduction. This yields the *VRU* update. Additionally, since $\theta^*$ and $\theta_r^*$ are close in parameter space (Lemma A.1), the low-variance gradient information anchored at $\boldsymbol{\theta}^*$ stays informative throughout the optimization trajectory. Unlike *SVRG*, no periodic recomputation is needed. This lead to faster convergence of *VRU* iterates, and thus faster unlearning, as quantified in the next subsection.

### 4.2. Convergence Speed of the *VRU* Algorithm

The following theorem characterizes the complexity of *VRU* in terms of the forget-set fraction $r_f$, the target error $e$, the privacy budget $\kappa_{\epsilon,\delta}$, and the loss geometry.

**Theorem 4.1.** *Let $\mathcal{F}$ be the set of $\mu$-strongly-convex, $L$-Lipschitz, and $\beta$-smooth loss functions. Then, for any $\ell \in \mathcal{F}$ and any $e > 0$, VRU achieves $(\varepsilon, \delta)-$unlearning and*

$$T_e(\ell, VRU) = \tilde{\mathcal{O}}\left(\kappa_\ell^3(1 + d\kappa_{\epsilon,\delta}^2 \log(\frac{1}{\delta}))\frac{e_0}{e}\left(\frac{r_f}{1-r_f}\right)^2\right). \tag{6}$$

**Proof sketch.** We start by upper-bounding the Lipschitz constant of $\tilde{\nabla}$ on a small ball around $\boldsymbol{\theta}^*$, then leverage this regularity to measure the speed of almost-sure convergence of the PSGD's final iterate to the global minimum $\boldsymbol{\theta}_r^*$, before applying the Gaussian mechanism (Dwork & Roth, 2014) to ensure unlearning. See App. A for the complete proof.

One highlight of this result is its quadratic dependence on the forget fraction, $\mathcal{O}(r_f^2)$, coupled with an $\mathcal{O}(1/e)$ dependence on the excess risk. The dependence on $r_f$ matches the best known rates for unlearning algorithms (Van Waerebeke et al., 2025), and the dependence in $e$ matches the lower bounds for stochastic optimization under our setup (Agarwal et al., 2009). Since $r_f$ is typically small, as low as $\mathcal{O}(1/n)$ for point-wise unlearning, this scaling translates into a substantial reduction in computational cost compared to full retraining or existing unlearning algorithms (see the next subsection). The result is stated in $\tilde{\mathcal{O}}$ rather than in $\mathcal{O}$ due to an additional factor in $\log\log T$ that arises when controlling the final iterate's distance to the global optimum $\boldsymbol{\theta}_r^*$ with high probability.

**Local condition number.** Although Theorem 4.1 is stated using the global condition number $\kappa_\ell = \beta/\mu$ for simplicity, the proof only uses smoothness and strong convexity inside the projection ball $B\left(\boldsymbol{\theta}^*, \frac{r_f}{1-r_f}\frac{L}{\mu}\right)$. Consequently, the relevant geometry is local around the trained model, not global over the entire loss landscape, providing a potentially smaller effective value of $\kappa_\ell$. This locality is further strengthened in the practical variant of *VRU*, where $L$ is replaced by $\|\nabla\mathcal{L}(\boldsymbol{\theta}^*, \mathcal{D}_f)\|$ (see Section 5.2).

### 4.3. Improvement Over Existing Methods

The best previously known convergence rate for $(\varepsilon, \delta)$-unlearning of functions in $\mathcal{F}$ is achieved by the "Noise and Fine-Tune" (*NFT*) algorithm (Neel et al., 2021). *NFT* also exhibits a quadratic dependence on the forget fraction, $\mathcal{O}(r_f^2)$ (Van Waerebeke et al., 2025), but has a worse dependence in $e$, namely $\mathcal{O}(1/e^2)$. This prevents *NFT* from outperforming retraining when a low excess risk is required. Theorem 4.1 establishes that *VRU* improves this dependence to $\mathcal{O}(1/e)$, meaning that the speedup *VRU* offers when compared to retraining does not decrease when dealing with smaller values of $e$. As a consequence, *VRU* has an improved bound over *NFT* for small values of $e$:

**Corollary 4.2.** *Let $e > 0$. Let $T_e^{max}(VRU)$ (resp. $T_e^{max}(NFT)$) be the best known asymptotical upper-bound on $T_e(VRU)$ (resp. $T_e(NFT)$). Then,*

$$\frac{T_e^{max}(VRU)}{T_e^{max}(NFT)} = \Theta\left(\kappa_\ell^3 \log(1/\delta)\frac{e}{e_0}\right). \tag{7}$$

Beyond improving upon existing unlearning methods, *VRU* also compares favorably to retraining when $r_f$ is small:

**Corollary 4.3.** *Under Assumption 3.2, let $0 < e < e_0$ and $\mathcal{A} \in \mathbb{A}$. Then,*

$$\frac{T_e(VRU)}{T_e(\mathcal{A})} = \tilde{\mathcal{O}}\left(\kappa_\ell^3\left(1 + d\kappa_{\epsilon,\delta}^2 \log\left(\frac{1}{\delta}\right)\right)\left(\frac{r_f}{1-r_f}\right)^2\right). \tag{8}$$

Corollary 4.3 addresses a natural question in machine unlearning: under what conditions can unlearning methods outperform retraining? The challenge lies in achieving favorable dependence on both the target excess risk $e$ and the forget fraction $r_f$ simultaneously. Retraining algorithms exhibit an $\mathcal{O}(1/e)$ dependence on excess risk, whereas *NFT*, despite its advantageous $\mathcal{O}(r_f^2)$ scaling, suffers from $\mathcal{O}(1/e^2)$. This means *NFT* loses its advantage over retraining in the low-error regime. By leveraging forget set gradients, *VRU* combines the $\mathcal{O}(r_f^2)$ scaling of unlearning methods with the $\mathcal{O}(1/e)$ dependence of retraining, substantially extending the regime of $(e, r_f)$ pairs for which unlearning offers meaningful computational gains.

### 4.4. Improvement Over Any Forget-Set-Free Method

In the following, we show the existence of a fundamental performance gap between *VRU* and *any* first-order forget-set-free $(\varepsilon, \delta)$-unlearning method.

A known limitation of first-order algorithms *without access to the forget set* is their inability to outperform retraining from scratch for a certain range of excess risks $e$ (Van Waerebeke et al., 2025). Crucially, *VRU* sidesteps this barrier by incorporating forget set gradients at each iteration. Theorem 4.4 formalizes the resulting separation.

As is standard in optimization complexity, and following existing work (Van Waerebeke et al., 2025), we assess an algorithm by its uniform performance over the function class $\mathcal{F}$. That is, rather than studying the convergence time of $\mathcal{U}$ on a fixed loss $\ell$, we consider its worst-case convergence time over $\mathcal{F}$:

$$T_e(\mathcal{U}) := \sup_{\ell \in \mathcal{F}} T_e\left(\ell, \mathcal{U}\right).$$

This quantity captures the number of iterations required by $\mathcal{U}$ to achieve accuracy $e$ for any admissible loss in the class.

With this quantity introduced, we are ready to showcase the performance gap between first-order $(\varepsilon, \delta)$-unlearning methods that do not leverage the forget set, and *VRU*.

**Theorem 4.4** (Fundamental gain from forget set access). *Assume Assumptions 3.1 and 3.2 hold. For any $\delta_{min} > 0$, there is a constant $c > 0$ such that, for any forget-set-free unlearning algorithm $\mathcal{U} \in \mathbb{U}_{\varepsilon,\delta}^r$, if $\delta \in [\delta_{min}, \varepsilon]$ and $e < c\kappa_{\epsilon,\delta}^2 \left(r_f/(1 - r_f)\right)^2 e_0$,*

$$\frac{T_e(\text{VRU})}{T_e(\mathcal{U})} = \tilde{\mathcal{O}}\left(d\kappa_{\epsilon,\delta}^2 \log\left(1/\delta\right)\kappa_\ell^3 \left(\frac{r_f}{1 - r_f}\right)^2\right).$$

**Corollary 4.5.** *Under the assumption of Theorem 4.4, for any forget-set-free unlearning algorithm $\mathcal{U} \in \mathbb{U}_{\varepsilon,\delta}^r$,*

$$\liminf_{r_f \to 0} \frac{T_e(\text{VRU})}{T_e(\mathcal{U})} = 0.$$

This theorem has important implications: it characterizes the complexity gain obtained by leveraging the forget set, by comparing the convergence rate of a specific forget-set–aware algorithm, *VRU*, with that of any algorithm that does not use the forget set. Therefore, in the error regime covered by the theorem, any first-order method that ignores the forget set is provably slower than *VRU* on the hardest losses in $\mathcal{F}$. This is a key advantage in typical unlearning use cases, where both $r_f$ and $e$ are small.

Regarding the regime of Theorem 4.4, the condition $e < c\kappa_{\epsilon,\delta}^2(r_f/(1 - r_f))^2 e_0$ should be interpreted as a sufficient low-error regime, not as a sharp characterization of all settings where forget-set access is useful. The theorem combines the upper bound of Theorem 4.1 with the forget-set-free lower bound of Van Waerebeke et al. (2025), and therefore inherits the constants and possible looseness of that lower bound. Importantly, it does not rule out empirical or instance-dependent gains outside this sufficient condition.

### 4.5. Discussion

Recent work by Mavrothalassitis et al. (2025) suggests that standard descent-ascent strategies (combining gradient ascent on $\mathcal{D}_f$ with gradient descent on $\mathcal{D}_r$) may degrade performance relative to the original model $\boldsymbol{\theta}^*$ and fail to converge to the optimum $\boldsymbol{\theta}_r^*$. However, the *VRU* algorithm lies outside the scope of this negative result due to structural differences: unlike the analyzed methods, *VRU* incorporates gradients computed at $\boldsymbol{\theta}^*$ and utilizes a three-terms update rule. More precisely, Mavrothalassitis et al. (2025) demonstrate that, for specific ranges of values of $r_f$ and $e$, descent-ascent cannot outperform retraining from scratch in regularized logistic regression settings. This stands in direct contrast to our experiment (see next section), and Corollary 4.3, which establishes values of $r_f$ for which *VRU* outperforms retraining, regardless of $e$.

Our theoretical analysis assumes that the unlearning procedure is initialized at the exact minimizer $\boldsymbol{\theta}^*$ of the full-data objective, as is standard in the certified unlearning literature. If the available model $\bar{\boldsymbol{\theta}}$ is only an approximate minimizer, two elements are affected in the convergence analysis. First, the projection ball in Alg. 1 may no longer be centered at the correct point; this can be handled by increasing its radius by an upper bound on $\|\bar{\boldsymbol{\theta}} - \boldsymbol{\theta}^*\|$, at the cost of larger constants in the subsequent analysis. Second, and more importantly, the variance-reduced estimator is no longer exactly unbiased. Indeed, anchoring the estimator at $\bar{\boldsymbol{\theta}}$ gives $\mathbb{E}[\tilde{\nabla}_t \mid \boldsymbol{\theta}_t] = \nabla\mathcal{L}_r(\boldsymbol{\theta}_t) - \frac{1}{1-r_f}\nabla\mathcal{L}(\bar{\boldsymbol{\theta}}, \mathcal{D})$, so the bias is controlled by the stationarity residual of the initial model. Under smoothness, this bias is at most $\beta\|\bar{\boldsymbol{\theta}} - \boldsymbol{\theta}^*\|/(1-r_f)$. Extending our proof to this setting would therefore require a high-probability convergence analysis for biased stochastic gradients; we leave this extension to future work.

# 5. Experiments

We empirically evaluate *VRU* against certified unlearning algorithms, empirical methods, and retraining baselines. Section 5.1 first describes all compared methods, and Section 5.2 details how to efficiently implement *VRU* in practice. Section 5.3 then compares our approach to certified methods and retraining baselines (Figure 1). Finally, Section 5.4 benchmarks against empirical unlearning algorithms, evaluating privacy leakage via membership inference attacks, as well as utility (Figure 2a). To compare *VRU* against other methods and assess the validity of our theory without introducing unnecessary complexity, we first consider a logistic regression task that ensures strong convexity. We extend our empirical comparison to a non-convex neural network in Section 5.5.

## 5.1. Compared Methods

To evaluate the performance of *VRU* relative to existing approaches, we compare with the $(\varepsilon, \delta)$ literature, the empirical one, and the retraining one. We choose the "Noise and Fine-Tune" (**NFT**) (Neel et al., 2021) algorithm as representative of the $(\varepsilon, \delta)-$unlearning methods. **NFT** achieves the best known utility-privacy tradeoff on losses in $\mathcal{F}$, as quantified in recent studies (Allouah et al., 2024b; Van Waerebeke et al., 2025). For empirical methods, we evaluate against: the **SCRUB** algorithm (Kurmanji et al., 2024), which alternates between maximizing the KL divergence on the forget set relative to the original model (the teacher) and minimizing the divergence from the teacher on the retain set, combined with a data fidelity term; the **NegGrad+** baseline (Kurmanji et al., 2024) that alternates between gradient ascent steps on $\mathcal{D}_f$ and descent steps on $\mathcal{D}_r$, always concluding with descent on $\mathcal{D}_r$ to preserve utility; the **Fine-Tune** baseline that only performs gradient descent on the retain set. Finally, we use the **GD**, **SGD** and **SVRG** (Johnson & Zhang, 2013) methods as retraining baselines.

In all experiments, methods are compared under an equal computational budget measured in *individual sample-gradient evaluations*, rather than in parameter updates or nominal epochs. This normalization accounts for different per-update costs: an algorithm that uses several gradients per update is run for proportionally fewer updates. For instance, since *VRU* requires multiple sample-gradient per update, it has a higher per-epoch computational cost than **NFT** and is therefore ran for fewer epochs.

## 5.2. Implementing the *VRU* Method in Practice.

When implementing *VRU*, several elements can be adapted to improve efficiency. While some conservative choices in Algorithm 1 ensure worst-case convergence guarantees, loss-specific adjustments can be made at run time without

compromising $(\varepsilon, \delta)$-unlearning. First, because the forget set is usually small, we can replace the stochastic forget gradient in Eq. 3 with a *full-batch* gradient $\nabla \mathcal{L}(\boldsymbol{\theta}^*, \mathcal{D}_f)$, computed once before optimization and then reused. This reduces the overall computational cost as soon as the unlearning computational budget exceeds roughly $r_f < 1$ fine-tuning epochs on the retain set. This substitution preserves unbiasedness and further reduces variance, since the full-batch gradient is deterministic. Second, we can use the forget set gradient's norm $\|\nabla \mathcal{L}(\boldsymbol{\theta}^*, \mathcal{D}_f)\|_2$ to replace the Lipschitz constant $L$, yielding the practical variant in Alg. 2, as proved in App. B. This is beneficial because $L$ is typically large and, for neural networks, can be NP-hard to compute (Virmaux & Scaman, 2018). Showing that the gradient norm suffices both broadens applicability and can improve convergence in practice.

Although the Lipschitz constant is not required to run *VRU*, it still needs the (local) strong convexity constant $\mu$. This dependence is not specific to *VRU*: certified first-order unlearning methods generally require curvature or sensitivity parameters to calibrate optimization and noise (Allouah et al., 2024b; Van Waerebeke et al., 2025). Misspecifying $\mu$ may affect the theoretical rate through the step-size schedule, but this is the same type of implementation issue encountered in standard strongly convex optimization and in other certified unlearning analyses, that might be alleviated by the use of modern parameter-free optimization tools (Robin et al., 2026).

Beyond these run-time adjustments that avoid reliance on $L$ and improve convergence speed, *VRU* has several built-in advantages over its empirical peers, particularly those relying on forget set gradient ascent. First, *VRU* does not require any hyperparameter to control the strength of gradient ascent. Empirical methods such as **SCRUB** or **NegGrad+** use biased gradient estimators: the optimum $\boldsymbol{\theta}_r^*$ does not represent a stationary point for them, as the average gradient on $\mathcal{D}_f$ is generally non-null at this point. Consequently, they must weigh their forget set gradient ascent with a carefully tuned rescaling factor in order to approximate the optimum, a time-consuming process. *VRU* sidesteps this issue entirely, as its gradient expectation is null at $\boldsymbol{\theta}_r^*$, and it has no tunable ascent hyperparameter. This stability guarantee enables convergence to $\boldsymbol{\theta}_r^*$ with arbitrarily high probability, unlike other ascent-based methods, which are known for their instability and potential for divergence. In contrast, running *VRU* longer always decreases the expected distance to the optimum.

Second, the only choice one must make when implementing *VRU* is the privacy budget $\kappa_{\epsilon, \delta}$, which can be tuned with negligible cost. A larger value of $\kappa_{\epsilon, \delta}$ yields stronger privacy guarantees but reduced utility. Importantly, this choice can be made *a posteriori*: because the noising step

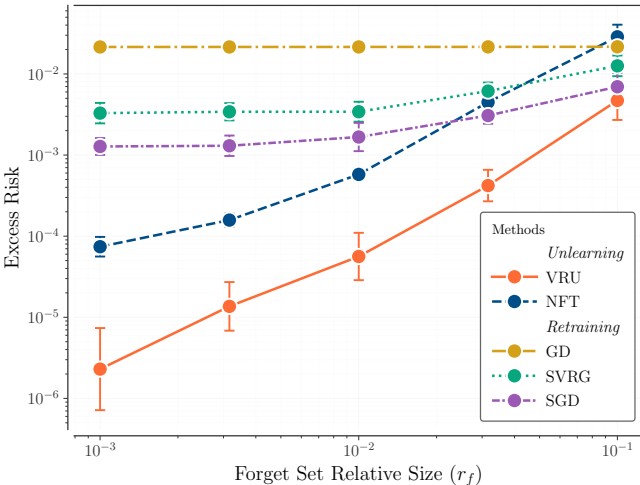

*Figure 1.* Excess risk of certified unlearning and retraining methods for varying forget fractions $r_f$, under fixed computational budget (10 epochs) and privacy budget ($\kappa_{\epsilon,\delta} = 1$). Results are averaged over 30 runs; error bars indicate $\pm 1$ standard deviation. *VRU* achieves the lowest excess risk across all tested $r_f$ values, with gains exceeding an order of magnitude for $r_f < 10^{-2}$.

is the final component of the algorithm, practitioners can decouple the optimization from the privacy decision and easily simulate several levels of noise without retraining, to find their application-specific sweet-spot. This contrasts with other noising-based approaches that often add noise before (Van Waerebeke et al., 2025) or during (Allouah et al., 2024b; Sepahvand et al., 2025) the optimization process.

### 5.3. Comparison to Certified Approaches

We first evaluate $(\varepsilon, \delta)$-unlearning methods alongside retraining baselines, which offer perfect privacy by training exclusively on $\mathcal{D}_r$. We consider a logistic regression task with cross-entropy loss and $\ell_2$ regularization to ensure strong convexity, using the Digit dataset (Alpaydin & Alimoglu, 1996). Full experimental details are provided in Appendix C.

**Setup.** We measure the excess risk achieved by each method across 5 values of $r_f$ spread logarithmically between $10^{-3}$ and $10^{-1}$, under a computational budget equivalent to 10 epochs of retraining from scratch with SGD and a privacy budget of $\kappa_{\epsilon,\delta} = 1$. For each trial, the forget set is selected uniformly at random and re-sampled across seeds. Results are averaged over 30 seeds; error bars in Figure 1 indicate $\pm 1$ standard deviation. Since computing the bounded sensitivity (Dwork & Roth, 2014) is intractable in most experimental settings, we measure it directly and provide it to all methods requiring it, ensuring privacy guarantees. Additional hyperparameters are provided in App. C.

**Results.** Figure 1 shows that for $r_f \leq 0.1$, *VRU* achieves a lower excess risk than all competing methods, with the

performance gap widening as $r_f$ decreases, reaching nearly two orders of magnitude at $r_f = 10^{-3}$. This behavior aligns with our theoretical predictions: *VRU* has better dependence on $e$ than **NFT**, making smaller values of $e$ attainable under a given computational constraint in the error regime described in Corr. 4.2.

### 5.4. Comparison to Empirical Approaches

Unlike certified methods, empirical unlearning algorithms lack formal privacy guarantees, necessitating empirical evaluation of privacy leakage. Following standard practice (Carlini et al., 2022; Hayes et al., 2024), we measure privacy risk via membership inference attacks (MIAs), which assess whether an adversary can distinguish forgotten samples from unseen test samples, a successful distinction indicating incomplete unlearning. Specifically, we implement U-LiRA (Hayes et al., 2024), the unlearning-adapted variant of the LiRA attack (Carlini et al., 2022). We report MIA accuracy as our measure of *empirical privacy risk*: an accuracy of 50% corresponds to random guessing (perfect unlearning), while higher values indicate privacy leakage. Implementation details are provided in Appendix C.

**Setup.** We evaluate all methods under a computational budget of 5 epochs across forget fractions $r_f \in \{3 \times 10^{-3}, 2 \times 10^{-2}, 10^{-1}\}$. For each method, we report both the excess risk $\mathcal{L}_r(\boldsymbol{\theta}) - \mathcal{L}_r(\boldsymbol{\theta}_r^*)$ and the empirical privacy risk (MIA accuracy). Results are averaged over 10 independent runs.

**Results.** Figure 2a presents the privacy-utility trade-off for each method, where the lower-left corner represents the ideal outcome (low excess risk, low privacy leakage).

For all tested values of $r_f$, *VRU* achieves the lowest empirical privacy risk among all methods. We observe that MIA accuracy remains close to 50% for most methods across all settings. This reflects a known limitation of membership inference attacks: they were developed primarily for complex, overparameterized models where memorization is prevalent (Carlini et al., 2022), and their discriminative power diminishes in strongly convex settings where all methods converge toward the unique optimum. Nevertheless, considerable differences appear. The **Fine-Tune** baseline exhibits the highest privacy leakage, particularly at small $r_f$. Unlike other methods, **Fine-Tune** lacks of mechanisms to actively degrade performance on the forget set, leaving its loss on forgotten samples low and thus vulnerable to attacks.

Regarding utility, *VRU* achieves the lowest excess risk for small and moderate values of $r_f$. At $r_f = 0.1$, however, *VRU* incurs higher excess risk than empirical methods as it is the only method to include a noise addition step, and the noise scales with $r_f$. This trade-off is expected: as $r_f$

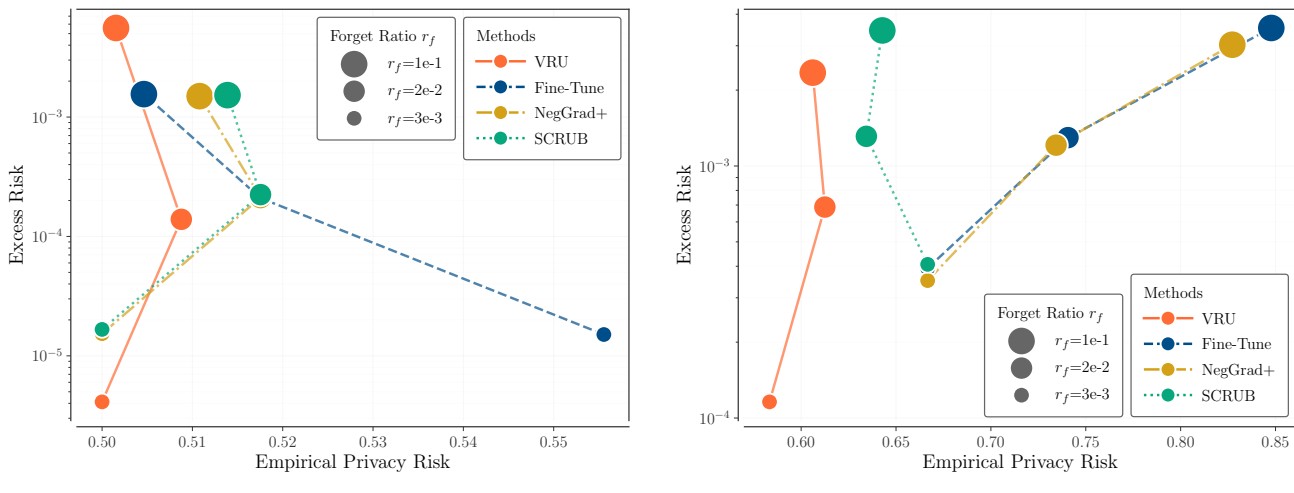

*(a)* Regularized logistic regression.   *(b)* Two-hidden-layer neural network.

*Figure 2.* Privacy–utility trade-off on the Digits dataset under a fixed computational budget. Each point corresponds to one method at a given forget fraction $r_f$. Left: regularized logistic regression. Right: two-hidden-layer neural network. **Excess risk** (y-axis): lower is better. **Empirical privacy risk** (x-axis): lower is better

increases, the computational advantage of unlearning over full retraining diminishes (cf. Figure 1), making $r_f \geq 0.1$ a less compelling regime for unlearning algorithms in general. In such cases, practitioners may prefer full retraining, which achieves both perfect privacy and comparable utility.

### 5.5. Non-Convex Evaluation

The theoretical analysis in this paper focuses on strongly convex, smooth, and Lipschitz losses, and the previous experiments were chosen to match this setting. To investigate whether *VRU* remains useful outside these assumptions, we also run a small neural-network experiment. We use the same dataset, but train a two-hidden-layer MLP with hidden widths 64 and 32, ReLU activations, and cross-entropy loss with $\ell_2$ regularization. This objective is non-convex and non-smooth.

**Setup.** Identical to the previous subsection (5.4). Since the assumptions of Theorem 4.1 are not satisfied in this setting, we do not claim a certified unlearning guarantee for this experiment.

**Results.** Figure 2b shows that *VRU* achieves a favorable privacy-utility trade-off in this small neural-network setting. In the plotted range, *VRU* out-competes the empirical baseline both in utility and privacy. This is encouraging, as the same update rule appears to behave well even when the loss no longer satisfies the assumptions used in our analysis. Larger experiments in deeper networks that show more aggressive non-convexity are yet to be performed.

## 6. Conclusion

In this work, we introduced Variance-Reduced Unlearning (*VRU*), the first first-order $(\varepsilon, \delta)$-unlearning algorithm to incorporate forget set gradients into its optimization process. By anchoring a variance-reduction mechanism at the pre-trained model, *VRU* bridges the gap between certified methods, which rely primarily on descent on the retain set, and empirical approaches, which also exploit the forget set but lack formal privacy guarantees.

Our convergence analysis for strongly convex, smooth, and Lipschitz losses establishes an $\mathcal{O}(1/e)$ dependence on the target excess risk, improving upon the $\mathcal{O}(1/e^2)$ scaling of prior certified methods and widening the regime where unlearning outperforms retraining. Beyond these improved rates, we proved a fundamental property: for a given range of error and forget ratio, *VRU* asymptotically outperforms any first-order $(\varepsilon, \delta)$-unlearning algorithm that does not access the forget set. Experiments inside and outside the scope of our theoretical analysis corroborate our theoretical findings, demonstrating consistent gains over both certified and empirical baselines.

Our analysis relies on strong convexity and, in principle, on the knowledge of the exact pre-trained optimum $\boldsymbol{\theta}^*$, though our experiments suggest robustness to inexact initialization. Additionally, while our bounds depend only on the local condition number, this dependence may still be restrictive in certain settings. A natural direction for future work is to extend *VRU* to relaxed notions of convexity that better capture the behavior of neural networks near local optima, such as the PŁ condition (Karimi et al., 2016; Liu et al., 2022) or the neural tangent kernel regime (Jacot et al., 2018).

## Impact Statement

This paper presents work whose goal is to advance the field of Machine Learning and Unlearning. There are many potential societal consequences of our work, none which we feel must be specifically highlighted here.

## Acknowledgements

This work was supported by the French government managed by the Agence Nationale de la Recherche (ANR) through France 2030 program with the references ANR-23-PEIA-005 (REDEEM), ANR-22-FAI1-0003 (TRAIN), ANR-24-IAS2-0001 (Fed-Ops), ANR-19-P3IA-0002 (3IA). This research was also supported in part by the European Network of Excellence dAIEDGE under Grant Agreement Nr. 101120726, by the EU HORIZON MSCA 2023 DN project FINALITY (G.A. 101168816), by the Groupe La Poste, sponsor of the Inria Foundation, in the framework of the FedMalin Inria Challenge.

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

# A. Proofs of main theoretical results.

We start by recalling this lemma from the literature, allowing to bound the distance between the starting point of our unlearning procedure and the global optimum.

**Lemma A.1** (Lemma C.2, Van Waerebeke et al. (2025))**.**

$$\|\boldsymbol{\theta}^* - \boldsymbol{\theta}_r^*\| \leq \frac{r_f}{1 - r_f} \cdot \frac{L}{\mu} \,. \tag{9}$$

We then bound the Lipschitz constant of *VRU*'s stochastic gradient estimator around $\boldsymbol{\theta}^*$, allowing for a better characterization of the local properties of the loss.

**Lemma A.2.** *For any* $\boldsymbol{\theta} \in B\left(\boldsymbol{\theta}^*, \frac{r_f}{1-r_f}\frac{L}{\mu}\right)$ *and* $t \in \mathbb{N}^*$,

$$\left\|\widetilde{\nabla}_t(\boldsymbol{\theta})\right\| \leq (1 + \kappa_\ell)\frac{r_f}{1 - r_f}L, \tag{10}$$

*where* $\kappa_\ell$ *is the condition number of the loss* $\ell$.

*Proof.*

$$\left\|\widetilde{\nabla}_t(\boldsymbol{\theta})\right\| \leq \left\|\nabla\ell(\boldsymbol{\theta}, \xi_r^t) - \nabla\ell(\boldsymbol{\theta}^*, \xi_r^t)\right\| + \left(\frac{r_f}{1 - r_f}\right)\left\|\nabla\ell(\boldsymbol{\theta}^*, \xi_f^t)\right\| \tag{11}$$

$$\leq \beta\left\|\boldsymbol{\theta} - \boldsymbol{\theta}^*\right\| + \left(\frac{r_f}{1 - r_f}\right)L \tag{12}$$

$$\underset{(1)}{\leq} \left(\frac{r_f}{1 - r_f}\right)L(1 + \kappa_\ell)\,, \tag{13}$$

where (1) is obtained through Lemma A.1. □

We are now in a position to prove the theorem.

**Theorem 4.1.** *Let* $\mathcal{F}$ *be the set of* $\mu$-*strongly-convex,* $L$-*Lipschitz, and* $\beta$-*smooth loss functions. Then, for any* $\ell \in \mathcal{F}$ *and any* $e > 0$, *VRU achieves* $(\varepsilon, \delta)$-*unlearning and*

$$T_e(\ell, \textit{VRU}) = \tilde{\mathcal{O}}\left(\kappa_\ell^3(1 + d\kappa_{\epsilon,\delta}^2\log(\frac{1}{\delta}))\frac{e_0}{e}\left(\frac{r_f}{1 - r_f}\right)^2\right)\,. \tag{6}$$

*Proof, Theorem 4.1.* Applying the *VRU* algorithm for $T$ iterations is equivalent to applying $T$ iterations of the PSGD algorithm (see *e.g.,* 12.4 in Garrigos & Gower (2023)) to the stochastic variance-reduced gradient $\widetilde{\nabla}_t$ with step size $1/\mu t$ then applying Gaussian noise with magnitude $\left(\frac{r_f}{1-r_f}\right)\nu_T\kappa_{\epsilon,\delta}$ (see Eq. 2). We remind that the stochastic gradient estimator $\widetilde{\nabla}$ is unbiased (Eq. 5). Using Lemma A.2, we bound the gradient of any sample with probability 1 and can thus apply Proposition 1 in Rakhlin et al. (2011). Hence, with probability at least $1 - \delta/2$,

$$\|\boldsymbol{\theta}_T - \boldsymbol{\theta}_r^*\|^2 \leq (624\log(2\log(T)/\delta) + 1)\left(\frac{r_f}{1 - r_f}\right)^2 L^2\frac{(1 + \kappa_\ell)^2}{\mu^2 T} \tag{14}$$

$$\leq \frac{1}{2}\left(\frac{r_f}{1 - r_f}\right)^2\nu_T^2 \tag{15}$$

where $\nu_T := \sqrt{2h(T,\delta)}L\frac{1+\kappa_\ell}{\mu\sqrt{T}}$ and $h(T,\delta) := 624\log(2\log(T)/\delta) + 1$.

This sensitivity bound is somewhat unusual: it only holds with high probability. We show in Lemma A.5 that applying an $(\varepsilon, \delta_1)$-DP mechanism when the sensitivity is bounded with probability $1 - \delta_2$ rather than deterministically achieves

$(\varepsilon, \delta_1 + \delta_2)-$DP. We apply this result with $\delta_1 = \delta_2 = \delta/2$. We thus define the noised estimate $\widetilde{\boldsymbol{\theta}}_T := \boldsymbol{\theta}_T + \left(\frac{r_f}{1-r_f}\right)\kappa_{\epsilon,\delta}\nu_T Z$, where $Z \sim \mathcal{N}(0,1)$, achieving $(\varepsilon,\delta)-$unlearning through the standard Gaussian mechanism (Dwork & Roth, 2014).

We can now evaluate the loss of the noised model:

$$\mathbb{E}\left[\mathcal{L}_r(\tilde{\boldsymbol{\theta}}_T) - \mathcal{L}_r(\boldsymbol{\theta}_r^*)\right] \leq \mathbb{E}\left[\mathcal{L}_r(\boldsymbol{\theta}_T) - \mathcal{L}_r(\boldsymbol{\theta}_r^*)\right] + \frac{\beta}{2}\mathbb{E}\left\|\tilde{\boldsymbol{\theta}}_T - \boldsymbol{\theta}_T\right\|^2 \tag{16}$$

$$\underset{(1)}{\leq} \frac{2}{T}\kappa_\ell e_0 \left(\frac{r_f}{1-r_f}\right)^2 (1+\kappa_\ell)^2 + \kappa_\ell(1+\kappa_\ell)^2 d\kappa_{\epsilon,\delta}^2 h(T,\delta)\left(\frac{r_f}{1-r_f}\right)^2 \frac{e_0}{2T} \tag{17}$$

$$\leq (1+\kappa_\ell)^2 \frac{e_0}{2T}\left(\frac{r_f}{1-r_f}\right)^2 \left(4\kappa_\ell + d\kappa_\ell\kappa_{\epsilon,\delta}^2 h(T,\delta)\right). \tag{18}$$

(1): The first term is obtained through Rakhlin et al. (2011)'s Theorem 1.

The inequality can be rewritten as $T \leq a + b\log\log T$, with $a > 0$ and $b > 0$. For $T \geq \exp(1)$, it holds

$$T \leq a + b\log\log T \leq a + b\log T \tag{19}$$
$$\leq (a+b)\log T. \tag{20}$$

From Lemma A.1 in Shalev-Shwartz & Ben-David (2014), it follows that

$$T \leq 2(a+b)\log(a+b). \tag{21}$$

We can then conclude:

$$T_e(\ell, \text{VRU}) = \tilde{\mathcal{O}}\left(\kappa_\ell^3(1 + d\kappa_{\epsilon,\delta}^2 \log(1/\delta))\frac{e_0}{e}\left(\frac{r_f}{1-r_f}\right)^2\right) \tag{22}$$

$\square$

Corollary 4.2 is obtained directly by dividing the result in Theorem 4.1 by the one obtained in Theorem 3 in Van Waerebeke et al. (2025).

Let $\mathcal{U} \in \mathbb{U}_{\varepsilon,\delta}^r$ be an unlearning algorithm that does not access the forget set gradient. Let $\mathcal{A} \in \mathbb{A}$ be retraining from scratch on the retain set $\mathcal{D}_r$ with the PSGD algorithm, as defined in Section 3 of Garrigos & Gower (2023).

We recall the next two results, describing the speed of $\mathcal{A}$ and comparing it to methods in $\mathbb{U}_{\varepsilon,\delta}^r$ and in $\mathbb{A}$.

**Theorem A.3** (Theorem 2, Van Waerebeke et al. (2025)). *Let $\delta \in [10^{-8}, \varepsilon]$. Under Assumption 3.2, for any $\delta_{min} > 0$, there exists a universal constant $c > 0$ such that, if $e < \min\left\{1, c\left(\frac{r_f}{1-r_f}\right)^2 \left(1 + \kappa_{\epsilon,\delta}^2\right)\right\}e_0$, then, for any $\mathcal{U} \in \mathbb{U}_{\varepsilon,\delta}^r$,*

$$\frac{T_e(\mathcal{U})}{T_e(\mathcal{A})} = \Omega(1). \tag{23}$$

**Lemma A.4** (Lemma 4.2, Van Waerebeke et al. (2025)). *Under Assumption 3.2, and if $e < e_0$, we have*

$$T_e(\mathcal{A}) = \Theta\left(\frac{e_0}{e}\right) \tag{24}$$

*Proof of Cor. 4.3:* We divide the upper bound in Theorem 4.1 by the lower bound in Lemma A.4 and the result follows. $\square$

We now have all the building blocks necessary to prove Theorem 4.4.

*Proof of Theorem 4.4.* Let $c$ be the constant in Theorem A.3. Let $e < c\kappa_{\epsilon,\delta}^2 \left(r_f/(1-r_f)\right)^2 e_0$ and $\delta \in [0, \epsilon]$. Let $\mathcal{U} \in \mathbb{U}_{\varepsilon,\delta}^r$. We control the speed of *VRU* compared to $\mathcal{A}$ through Cor. 4.3. We control the speed of $\mathcal{U}$ compared to $\mathcal{A}$ through Theorem A.3. Then,

$$\frac{T_e(\text{VRU})}{T_e(\mathcal{U})} = \frac{T_e(\text{VRU})}{T_e(\mathcal{A})}\frac{T_e(\mathcal{A})}{T_e(\mathcal{U})} = \tilde{\mathcal{O}}\left(\left(1 + d\kappa_{\epsilon,\delta}^2 \log\left(\frac{1}{\delta}\right)\right)\kappa_\ell^3\left(\frac{r_f}{1-r_f}\right)^2\right). \tag{25}$$

$\square$

---

**Algorithm 2 VRU-exp** (Variance Reduced Unlearning, experiments version)

---

**Require:** Number of iterations $T$, trained model $\boldsymbol{\theta}^*$, loss function $\ell$, forget ratio $r_f \in (0, 1)$, retain set $\mathcal{D}_r$, forget set $\mathcal{D}_f$, privacy budget $\kappa_{\epsilon, \delta}$

1: Set $\theta_0 = \theta^*$,
2: Compute a full-batch gradient on the forget set: $\nabla_f^* := \nabla \mathcal{L}(\boldsymbol{\theta}^*, \mathcal{D}_f)$
3: Compute the improved error value $\nu_T^{\text{exp}}$ (Eq. 32).
4: Set $R^{\text{exp}} := \frac{r_f}{1-r_f} \frac{\|\nabla_f^*\|}{\mu}$
5: **for** $t = 0$ to $T - 1$ **do**
6:     Sample data point: $\xi_t^r \sim \mathcal{D}_r$,
7:     Compute variance-reduced gradient:

$$\tilde{\nabla}_t \leftarrow \nabla \ell(\boldsymbol{\theta}_t, \xi_t^r) - \nabla \ell(\boldsymbol{\theta}^*, \xi_t^r) - \frac{r_f}{1 - r_f} \nabla_f^*$$

8:     Update parameters and perform projection on the ball: $\boldsymbol{\theta}_{t+1} \leftarrow \text{proj}_{B(\boldsymbol{\theta}^*, R^{\text{exp}})} \left( \boldsymbol{\theta}_t - 1/\mu t \, \tilde{\nabla}_t \right)$
9: **end for**
10: Sample $Z \sim \mathcal{N}(0, 1)$
11: Noise the model to ensure unlearning:

$$\tilde{\boldsymbol{\theta}}_T = \boldsymbol{\theta}_T + \left( \frac{r_f}{1 - r_f} \right) \nu_T^{\text{exp}} \kappa_{\epsilon, \delta} Z$$

12: **return** Final model $\tilde{\boldsymbol{\theta}}_T$

---

**Technical lemma.** We show how a high-probability bound on the sensitivity can still translate to $(\varepsilon, \delta)-$differential privacy by adding the failure probabilities of the bound and the DP. While we are probably not the first to prove this result, we were unable to find a direct formulation elsewhere.

**Lemma A.5** (Differential Privacy under High-Probability Sensitivity). *Suppose that a sensitivity bound $\Delta$ holds with probability at least $1 - \delta_1$. If a mechanism $\mathcal{M}$ satisfies $(\varepsilon, \delta_2)$-DP when the sensitivity is at most $\Delta$, then $\mathcal{M}$ satisfies $(\varepsilon, \delta_1 + \delta_2)$-DP.*

*Proof.* Let $E$ be the event that the sensitivity bound holds, so $\Pr[E] \geq 1 - \delta_1$. For any measurable $S$:

$$\begin{aligned}
\Pr[\mathcal{M}(D) \in S] &= \Pr[\mathcal{M}(D) \in S \mid E] \Pr[E] + \Pr[\mathcal{M}(D) \in S \mid E^c] \Pr[E^c] \\
&\leq \left( e^\varepsilon \Pr[\mathcal{M}(D') \in S \mid E] + \delta_2 \right) \Pr[E] + \Pr[E^c] \\
&\leq e^\varepsilon \Pr[\mathcal{M}(D') \in S] + \delta_1 + \delta_2. \qquad \square
\end{aligned}$$

# B. Run-time improvements of *VRU*

We introduce the following results with practical implementation of in mind. They providing guidance for implementing *VRU* more efficiently while preserving its theoretical guarantees. In particular, Alg. 2 describes a practical implementation of *VRU* that avoids requiring the Lipschitz constant $L$, which is often intractable to compute in practical settings. The convergence speed of Alg. 2 to $\boldsymbol{\theta}_r^*$ can be proven by replacing Lemma A.1 by Lemma B.1, and Lemma A.2 by Lemma B.2 in the proof of Theorem 4.1. The update rule for Alg. 2 thus becomes

$$\widetilde{\nabla}^{\text{exp}}(\boldsymbol{\theta}, \xi^r) = \nabla \ell(\boldsymbol{\theta}, \xi^r) - \nabla \ell(\boldsymbol{\theta}^*, \xi^r) - \frac{r_f}{1 - r_f} \nabla \mathcal{L}(\boldsymbol{\theta}^*, \mathcal{D}_f), \tag{26}$$

where the full-batch gradient on $\mathcal{D}_f$ is only computed once, before optimization begins.

The following result is similar to Lemma A.1, but leverages specific forget set gradient, not the worst-case Lipschitz bound,

**Lemma B.1** (Bounded optima distance).

$$\|\boldsymbol{\theta}^* - \boldsymbol{\theta}_r^*\| \leq \frac{r_f}{1 - r_f} \cdot \frac{\|\nabla \mathcal{L}(\boldsymbol{\theta}^*, \mathcal{D}_f)\|}{\mu}. \tag{27}$$

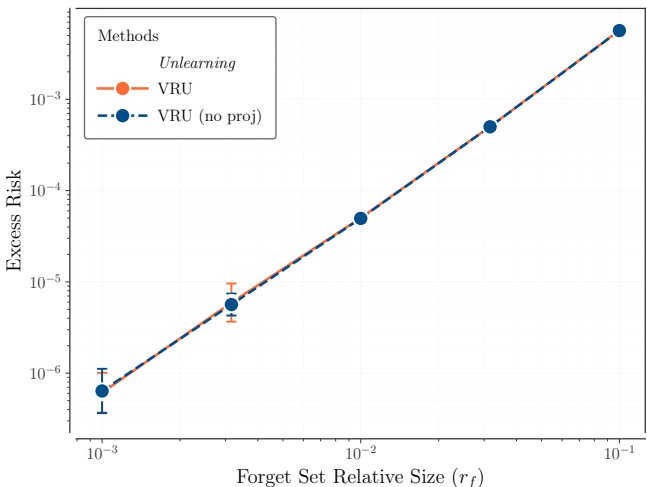

*Figure 3.* Ablation study on the projection step. Excess risk versus forget fraction $r_f$ for *VRU* with and without projection, using $\kappa_{\epsilon,\delta} = 0.1$. The projection step has minimal impact on convergence, indicating *VRU*'s robustness to this algorithmic choice. Error bars: $\pm 1$ std. over 30 runs.

*Proof.* By strong convexity of $\mathcal{L}(\cdot, \mathcal{D}_r)$, we have $\mu \|\boldsymbol{\theta}^* - \boldsymbol{\theta}_r^*\| \leq \|\nabla \mathcal{L}_r(\boldsymbol{\theta}^*)\|$. Additionally, $\|\nabla \mathcal{L}(\boldsymbol{\theta}^*, \mathcal{D}_r)\| = \frac{r_f}{1-r_f} \|\nabla \mathcal{L}(\boldsymbol{\theta}^*, \mathcal{D}_f)\|$. This concludes the proof. □

Instead of bounding $\left\|\widetilde{\nabla}_t(\boldsymbol{\theta})\right\|$ on $B\left(\boldsymbol{\theta}^*, \frac{r_f}{1-r_f}\frac{L}{\mu}\right)$, we can reduce the radius of the ball to $B\left(\boldsymbol{\theta}^*, \frac{r_f}{1-r_f}\frac{\|\nabla \mathcal{L}(\boldsymbol{\theta}^*, \mathcal{D}_f)\|}{\mu}\right)$ (see Lemma B.1). This also allows for a reduction in the gradient's bound, as described in the following result.

**Lemma B.2.** *For any* $\boldsymbol{\theta} \in B\left(\boldsymbol{\theta}^*, \frac{r_f}{1-r_f}\frac{\|\nabla \mathcal{L}(\boldsymbol{\theta}^*, \mathcal{D}_f)\|}{\mu}\right)$ *and* $t \in \mathbb{N}^*$,

$$\left\|\widetilde{\nabla}_t(\boldsymbol{\theta})\right\| \leq \frac{r_f}{1-r_f}(1+\kappa_\ell)\|\nabla \mathcal{L}(\boldsymbol{\theta}^*, \mathcal{D}_f)\| \ , \tag{28}$$

*where* $\kappa_\ell$ *is the condition number of the loss l, and the supremum is defined as the set is non-empty and upper-bounded by L.*

*Proof.*

$$\left\|\widetilde{\nabla}_t(\boldsymbol{\theta})\right\| \leq \left\|\nabla \ell(\boldsymbol{\theta}, \xi_r^t) - \nabla \ell(\boldsymbol{\theta}^*, \xi_r^t)\right\| + \left(\frac{r_f}{1-r_f}\right)\|\nabla \mathcal{L}(\boldsymbol{\theta}^*, \mathcal{D}_f)\| \tag{29}$$

$$\leq \beta \|\boldsymbol{\theta} - \boldsymbol{\theta}^*\| + \left(\frac{r_f}{1-r_f}\right)\|\nabla \mathcal{L}(\boldsymbol{\theta}^*, \mathcal{D}_f)\| \tag{30}$$

$$\underset{(2)}{\leq} \frac{r_f}{1-r_f}(1+\kappa_\ell)\|\nabla \mathcal{L}(\boldsymbol{\theta}^*, \mathcal{D}_f)\| \ , \tag{31}$$

where (2) is obtained through Lemma B.1. □

To simplify the expression of the VRU empirical algorithm, we define

$$\nu_T^{\text{exp}} := \frac{\sqrt{2h(T,\delta)}}{\mu\sqrt{T}}\|\nabla \mathcal{L}(\boldsymbol{\theta}^*, \mathcal{D}_f)\|(1+\kappa_\ell). \tag{32}$$

## C. Implementation Details

### C.1. Common Experimental Setup

**Dataset and Model.** All experiments in the main manuscript use the Digits dataset (Alpaydin & Alimoglu, 1996) with either a logistic regression model, or a two-hidden-layer neural network, trained using cross-entropy loss, coupled with a $L2$-weight regularization of weight 0.1. Training uses a batch size of 8.

*Table 1.* Hyperparameters for each unlearning and retraining method in section 5.3.

| Method | Learning Rate | LR Decay | Method Type |
|--------|---------------|----------|-------------|
| VRU | 1.1 | 0.55 | Unlearning |
| NFT | $3 \times 10^{-1}$ | 0.8 | Unlearning |
| GD | 2.0 | 0.8 | Retraining |
| SVRG | 1.0 | 0.4 | Retraining |
| SGD | 0.5 | 0.9 | Retraining |

*Table 2.* Hyperparameters for each unlearning method in section 5.4.

| Method | Learning Rate | LR Decay | $\alpha$ |
|--------|---------------|----------|----------|
| VRU | 1 | 0.6 | – |
| Fine-Tune | $5 \times 10^{-3}$ | 0.8 | – |
| NegGrad+ | $3 \times 10^{-3}$ | 0.7 | $5 \times 10^{-3}$ |
| SCRUB | $5 \times 10^{-3}$ | 0.8 | $5 \times 10^{-3}$ |

**Evaluation Protocol.** Excess risk measures the gap between the unlearned model's test loss and the retrained-from-scratch baseline. The results are aggregated across seeds, reporting means with standard errors. Geometric means and error bars are used as the measured variables tend to span across several orders of magnitude.

### C.2. Experimental Setup for Figure 1

**Unlearning Configuration.** We set the unlearning epoch budget to $T = 10$ and evaluate $n_{r_f} = 5$ uniformly spaced forget ratios in the range $r_f \in [10^{-3}, 0.1]$. We run each experiment with 30 independent random seeds: 0 through 29.

**Method-Specific Hyperparameters.** Table 1 summarizes the hyperparameters for each unlearning method.

### C.3. Experimental Setup for Figure 2a

**Unlearning Configuration.** We set the unlearning epoch budget to $T = 5$, as empirical methods target steeper computational gains, and evaluate three forget ratios: $r_f \in \{3 \times 10^{-3}, 2 \times 10^{-2}, 10^{-1}\}$. We do not take $r_f$ smaller than $3 \times 10^{-3}$ as results become too unstable when attacking only a few samples, *i.e.,* only 0%, 50%, or 100% accuracy for any attack on $|\mathcal{D}_f| = 1$. We run each experiment with 10 random seeds.

**Method-Specific Hyperparameters.** Table 2 summarizes the hyperparameters for each unlearning method. The learning rate decays by the specified factor after each epoch. The parameter $\alpha$ denotes the weight of the ascent step in SCRUB and NegGrad+.

As this is an empirical evaluation, unlike for the previous subsection, we do not provide the bounded sensitivity to methods requiring it. For *VRU*, the noise is applied empirically with $\kappa_{\epsilon,\delta} = 0.1$, which offers a good trade-off, taking $\nu_T = 1$, as the smoothness parameter is not known. However, knowing the value of $\nu_T$ would not change our results, as it would simply re-scale the noise, which is equivalent to re-scaling $\kappa_{\epsilon,\delta}$, whose value is not relevant, nor reported in the main text, in this comparison to empirical methods, that do not offer any level of $(\varepsilon, \delta)-$unlearning.

**Privacy Evaluation via Membership Inference.** We assess privacy risk using the U-LiRA membership inference attack with 5 shadow models. Target shadow models are trained on $\mathcal{D} = \mathcal{D}_f \cup \mathcal{D}_r$ and then unlearn $\mathcal{D}_f$, while reference shadow models are trained from scratch on $\mathcal{D}_r$ only. To construct the attack set, we sample $n_f = |\mathcal{D}_f|$ elements uniformly at random from the test set to form $\mathcal{D}_{\text{test}}$. The attack set is $\mathcal{D}_{\text{attack}} = \mathcal{D}_f \cup \mathcal{D}_{\text{test}}$, and we report the U-LiRA re-identification attack accuracy as the empirical privacy risk.

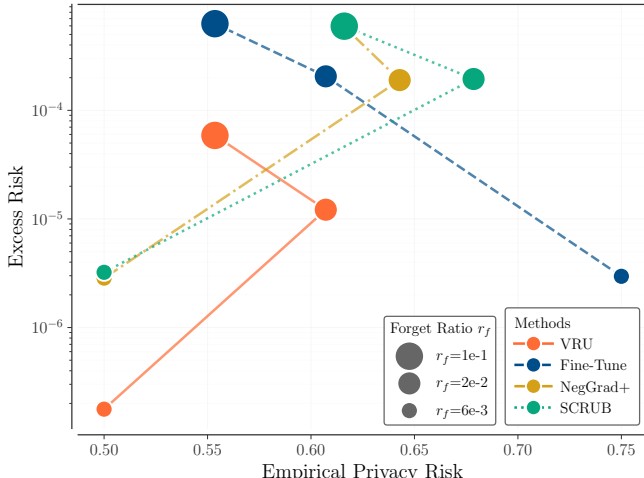

*Figure 4.* Privacy–utility trade-off for regularized logistic regression on the Breast Cancer Wisconsin dataset under a fixed computational budget of 5 epochs. Each point corresponds to one method at a given forget fraction $r_f$. **Excess risk** (y-axis): lower is better. **Empirical privacy risk** (x-axis): U-LiRA accuracy, lower is better. The lower-left region represents the best trade-off.

## C.4. Impact of the projection step

The *VRU* algorithm, even in its experiment-adjusted form (Alg. 2), requires a projection step on a ball of radius $R^{\exp} :=$ $\frac{r_f}{1-r_f} \frac{\|\nabla_f^*\|}{\mu}$ after each iteration. If this ball's radius was to be incomputable, one might wonder if the algorithm would still function properly. We answer this question by analyzing the performance of *VRU* without the projection step. We place ourself in the experimental setting of Figure 2a, and report the result in Figure 3, showing the excess risk as a function of $r_f$, for 5 logarithmically-spread values of in $[10^{-3}, 0.1]$. While we remind that removing the projection step loses the unlearning certification of the algorithm, we find that the projection step has little to no impact on the loss evolution and is not necessary for the algorithm to effectively reach the optimum in our experimental setting. The reported error bars represent $\pm 1$ standard deviations over 30 independent runs.

## D. Additional Logistic Regression Experiment on Breast Cancer

We report an additional experiment on the Breast Cancer Wisconsin dataset, using the implementation provided by `sklearn.datasets.load_breast_cancer`. The dataset contains 569 samples, each represented by 30 real-valued features, and two classes. Inputs are standardized before training. As in the main Digits experiment, we use a regularized logistic regression model trained with cross-entropy loss and $\ell_2$ regularization, and we compare *VRU* against **Fine-Tune**, **NegGrad+**, and **SCRUB** under a fixed computational budget.

The protocol follows the one used in the main manuscript for the Digits logistic-regression experiment. We run each method for $T = 5$ epochs with batch size 64 and weight decay 0.1, and evaluate three forget fractions, chosen to span values of $r_f$ from a handful of forget sample to a sizable portion of the dataset,

$$r_f \in \{6 \cdot 10^{-3}, 2.45 \cdot 10^{-2}, 10^{-1}\}.$$

Privacy leakage is measured using the same U-LiRA evaluation protocol as in the neural-network experiment, with Gaussian likelihood-ratio tests based on true-class logit scores. Results are averaged over two random seeds.

**Method-Specific Hyperparameters.** Table 3 summarizes the hyperparameters for each unlearning method in the Breast Cancer Wisconsin experiment. The learning rate decays by the specified factor after each epoch. The parameter $\alpha$ denotes the weight of the ascent step in SCRUB and NegGrad+.

Figure 4 shows that the conclusions from the Digits experiment also hold on this additional dataset. *VRU* achieves the lowest excess risk among the tested methods across the considered forget fractions, while maintaining competitive empirical privacy risk. The improvement is most pronounced for small forget fractions, where the variance-reduced use of the forget-set gradient leads to substantially lower excess risk than the empirical baselines under the same computational budget. This

*Table 3.* Hyperparameters for each unlearning method in the Breast Cancer Wisconsin experiment.

| Method | Learning Rate | LR Decay | $\alpha$ |
|---|---|---|---|
| VRU | 1 | 0.7 | – |
| Fine-Tune | $1 \times 10^{-3}$ | 0.7 | – |
| NegGrad+ | $5 \times 10^{-3}$ | 0.7 | $2 \times 10^{-2}$ |
| SCRUB | $5 \times 10^{-3}$ | 0.8 | $1 \times 10^{-2}$ |

provides additional evidence that the behavior observed in the main logistic-regression experiment is not specific to the Digits dataset.

