# OpenReview forum: "Variance-Reduced $(\varepsilon, \delta)-$Unlearning using Forget Set Gradients"
_ICML.cc/2026/Conference — ICML 2026 regular_

### Official Review · Reviewer_T71M · 2026-03-06

**Soundness:** 2
**Presentation:** 3
**Significance:** 2
**Originality:** 3
**Overall Recommendation:** 4
**Confidence:** 3

**Summary:**

The paper studies optimization for the problem of unlearning. The data distribution is a mixture of the retaining set and the forget set $\mathcal{D} = (1-r_f)\mathcal{D_r} +r_f \mathcal{D_f}$. The authors propose a new gradient algorithm that exploits the forget set in the gradient update using variance reduction techniques. Starting from the trained model, $\theta_0 = \theta^\*$, the gradient update is given by $\nabla = \nabla f(\theta_t, \xi_r) -  \nabla f(\theta^\*, \xi_r) - \frac{r_f}{1-r_f}  \nabla f(\theta^*, \xi_f)$. This algorithm improves the convergence rate from $1/e^2$ for fine-tuning algorithm (NFT) to $1/e$ and improves the convergence for forget-set-free algorithms (including retraining on the retaining set) when $r_f \ll 1$ and when the error $e$ is small. The algorithm is empirically validated on the Digit dataset where its performance is compared against several algorithms in terms of excess risk and privacy risk.

**Compliance With Llm Reviewing Policy:**

Affirmed.

**Final Justification:**

My concerns were addressed. I would recommend the authors to include the discussion here to clarify the contribution of the paper.

**Key Questions For Authors:**

- Can you check Eq (17)?
- In Figure 1, since VRU uses at least 2 gradient computations in each step (since the implementation uses a fix gradient for the forget set), how do you compare with non VR methods fairly? Shouldn't we use twice the number of epochs for NFT and SGD?
- In Figure 3, I don't think VRU with no projection should be considered at all because we cannot guarantee the privacy.

**Limitations:**

- Limitations were discussed but some key points are somewhat lacking.

**Strengths And Weaknesses:**

**Strengths**
- The paper is well motivated and the result is positive: using the forget set in unlearning can improve the quality.
- The main contribution of this paper is the new variance-reduced algorithm which has an improved convergence rate in terms of the target error. Variance reduction has been used successfully in optimization, but I find that the way it is exploited in this paper is quite different and interesting. The anchor is always $\theta^\*$, and samples from the forget set is used because we know that the gradient at $\theta^*$ is 0, which means we have an unbiased gradient estimate.
- The presentation of the paper is mostly clear and very easy to follow.

**Weaknesses**
- The issue I find with the proof of Theorem 4.1 is in inequality (17) (Line 665). I believe why the error $e_0$ appears multiplicatively in the bound is only when the step size is chosen to have the error term. But the algorithm set the step size to be $1/\mu t$ so the bound is not what is stated in (17). Furthermore, In the same line, there should be a term $L^2$ (any maybe another $1/\mu$). So there is an issue with the rate in Theorem 4.1, where there should be an extra factor $L^2$ at least.
- Regarding the convergence rate, the dependence on the condition number $\kappa_\ell^3$ seems too high. For problems with a large condition number this is problematic. This doesn't seem to be properly discussed in the paper.
- The regime for which there is a fundamental improvement over forget-set-free method is also limited, it requires the error to be small, and also there is a dependence $\kappa_\ell^3$.
- The only main technical contribution is only Theorem 4.1, the other results are straight from the prior work by Van Waerebeke et al, so I feel here it is somewhat limited.
- Experiments seem quite limited as well, only one dataset is considered.
- Even though it might seem obvious from the algorithm description, one point that I think should be made more explicit in the theorem statement is that VRU satisfies $(\epsilon,\delta)$-unlearning. Why VRU satisfies $(\epsilon,\delta)$-unlearning only appears in the proof of Theorem 4.1 in the appendix.

---

> ### Author Rebuttal · Authors · 2026-03-30
>
> **Eq. $17$ concern.** We deeply appreciate the attention to proof details in the appendix shown by Reviewer T71M. Their comment stems from valid grounds, as we forgot to introduce the quantity $e_0:=\frac{L^2}{\mu}$. We apologize for this oversight. $e_0$ is, up to a factor 4, the largest excess risk achievable by a model, and is thus used to bound the error made at initialization. Substituting $e_0$ for $\frac{L^2}{\mu}$ in Eq. $(17)$ solves both the issues raised by the reviewer.
>
> **Fair comparison between methods.** It is indeed necessary to compare methods fairly. We confirm that *NFT* and *SGD* do train for more than double the epochs of *VRU* (see L318-323, right). We discuss this in further detail in our response to reviewer fSLX.
>
> **Theorem 4.4's regime of application.** The reviewer raises an interesting point that we will discuss more clearly in our manuscript: the condition  $e<c \kappa^2_{\epsilon,\delta}(r_f/(1-r_f))^2e_0$ can indeed be restrictive and deserves a more precise discussion.
> First, we emphasize that this is a sufficient condition for Thm. 4.4 to hold, but not a necessary one. In particular, Van Waerebeke et al. (2025) suggest that there might be a missing factor "$d$" in this inequality on the right-side, a large discrepancy in high dimension. Moreover, their constant $c$ is derived through a long sequence of inequalities and may therefore be quite loose. Despite these limitations, one can still evaluate the threshold given by Thm. 4.4, keeping in mind that it may be excessively pessimistic. With $c\approx 5 \times 10^{-6}$, $r_f=0.01$, a reasonably strict unlearning budget of $\epsilon=1$ and $\delta=10^{-6}$, $L=100$, $\mu=0.1$, this gives $e<c\kappa^2_{\epsilon,\delta} (rf/(1-rf))^2e_0\approx10^{-4}$, which corresponds to a low but still relevant error regime, as illustrated in Figs. $1$ and $2$.
>
> **Impact of $\kappa_\ell$.** The reviewer rightly identifies the condition number as a factor limiting the *VRU* convergence speed. However, $\kappa_\ell$ reflects the *local* geometry of the loss rather than its global landscape (L262-267, left). In particular, *VRU* keeps the iterates within a small ball around the initialization point, whose radius scales with $r_f$. As a result, the curvature variations within this region are smaller than those over the full landscape, which in turn yields a smaller $\kappa_\ell$. We will clarify this distinction in the manuscript.
>
> **Limited main theoretical contribution.** We agree that Thm. 4.1 is the central new technical ingredient of the paper. However, we respectfully disagree that the paper’s technical contribution is therefore limited to that theorem alone. A main contribution is the introduction of *VRU*, the first first-order $(\epsilon,\delta)$-unlearning method that directly uses forget-set gradients through a variance-reduced estimator, together with its convergence analysis. Thm. 4.1 is the key positive result establishing the new $\tilde{O}(r_f^2/e)$ rate; this qualitatively improves over the previous $\tilde{O}(r_f^2/e^2)$ dependence discussed in the paper and is precisely what enables the low-error regime where certified unlearning remains faster than prior certified methods and can also beat retraining.
> Regarding Thm. 4.4 and Cor. 4.3: we agree their proofs combine our upper bound with lower bounds from Van Waerebeke et al. (2025) and will make this dependency more explicit. However, Van Waerebeke et al. do not provide a method using forget-set gradients, nor establish a separation between such methods and forget-set-free first-order certified unlearning algorithms. This separation only becomes possible once Theorem 4.1 is proved, making these results new and important consequences, regardless of their proof technique.
> More broadly, the paper’s contribution is not only a single bound but a new algorithmic and conceptual message: incorporating forget-set gradients can provably improve first-order certified unlearning. We will revise the presentation to better distinguish the core new theorem from the downstream consequences obtained by combining it with prior lower bounds.
>
> **Limited experimental scope.** We broaden our empirical validation by including experiments on another dataset, and with a neural network, as detailed in our response to Reviewer fSLX.
>
> **Unlearning clarification for Theorem 4.1.** We thank the reviewer for this helpful suggestion. While the unlearning guarantee is indeed achieved through the added noise, the current formulation of Thm. 4.1 does not make this sufficiently explicit. We will add "*VRU* achieves $(\epsilon,\delta)$-Unlearning, and" before Eq. $6$.
>
> **Relevance of Figure 3.** The reviewer is right that *VRU* without the projection step lacks unlearning guarantees. We included this comparison in the Appendix only for completeness, to assess whether the method remains practically relevant when the projection ball radius, which depends on $L$, $\mu$, and $r_f$, cannot be computed.

---

> > ### Author Rebuttal · Reviewer_T71M · 2026-04-02
> >
> > I still think more work is needed to address the concerns about the convergence and regime of applications, but I'm happy to raise the score.

---

### Official Review · Reviewer_fSLX · 2026-03-13

**Soundness:** 3
**Presentation:** 4
**Significance:** 2
**Originality:** 3
**Overall Recommendation:** 4
**Confidence:** 2

**Summary:**

This paper studies certified machine unlearning under the
(ε,δ)-unlearning framework. The main idea is to use forget-set gradients directly during optimization, instead of using the forget set only to calibrate noise as in prior first-order certified methods. The proposed method, VRU, is presented as the first first-order certified unlearning algorithm to do this, with theory showing improved convergence rates and a separation from first-order methods that do not access the forget set. Empirically, the paper evaluates on strongly convex logistic regression and shows better excess risk than certified baselines and favorable privacy-utility tradeoffs against empirical baselines, especially when the forget ratio is small.

**Compliance With Llm Reviewing Policy:**

Affirmed.

**Final Justification:**

I keep my original score.

**Key Questions For Authors:**

Please address the weakness section.

**Limitations:**

yes

**Strengths And Weaknesses:**

Strengths:

The main strength of the paper is its meaningful theoretical contribution. Although I did not examine the proofs in full detail, the arguments appeared logical and well motivated. I also appreciated the overall writing quality of the paper, particularly its clear flow and the way it positions itself within the literature through a well-organized related work section. The paper is well situated within the area of certified first-order unlearning, and its novelty is clear: VRU directly incorporates forget-set gradients while still preserving certified guarantees. The theoretical improvement from the prior $O(1/\epsilon^2)$ dependence to $O(1/\epsilon)$, together with the result that VRU can asymptotically outperform any first-order certified method that does not use forget-set gradients, makes the contribution feel substantial rather than incremental.

Weaknesses:

A major weakness is that the paper is limited to a narrow setting. Both the theory and experiments rely on strongly convex, smooth, and Lipschitz objectives, and the experiments are carried out on logistic regression to match that setting. As a result, the paper’s practical relevance to modern deep neural networks remains unclear.

Another weakness is that the empirical advantage is somewhat qualified. VRU has a higher per-epoch cost than NFT, so its benefit comes more from making better progress under a fixed compute budget than from being a simpler or cheaper method. Moreover, at larger forget ratios, the added certification noise degrades utility, and the paper itself notes that full retraining may then be preferable. As a result, the method appears most convincing in the small-forget regime, rather than as a broadly dominant unlearning approach.

---

> ### Author Rebuttal · Authors · 2026-03-30
>
> We thank reviewer fSLX for recognizing the originality of our theoretical contributions and the clarity of our writing. We address the weaknesses they raised below.
>
> **Fair comparison between methods.** The reviewer is right to assess that *VRU* has a higher per-epoch cost than *NFT*. However, as stated in L318-323 (right), all our experiments are conducted under a strictly identical computational budget across methods. Accordingly, when *NFT* performs $10$ fine-tuning epochs, *VRU* performs slightly less than $5$ (as it requires an initial full-batch forget set gradient computation and then uses $2$ gradient calls per sample), yet it still largely outperforms it. Other methods also have their running time adapted to their per-iteration cost: *NegGrad+* uses $2$ gradient calls per sample and is thus run for half the number of Fine-Tune's epochs, $SVRG$ uses a total of $3$ full-batch gradients on the retain set during its trajectory, as well as $2$ gradients per sample, and is thus run for $(10-3)/2=3.5$ epochs.
> The evolution of the loss curves over iterations shows that, in our experimental setting, VRU remains more computationally efficient than NFT across the range of target errors $e$ considered. We will make this point clearer in the revised manuscript and can provide additional curves if needed.
>
> **Assumptions and non-convex experimental validation.**
> The reviewer is right to point out that our theoretical analysis relies on strong assumptions. We would first emphasize, however, that such assumptions are needed to derive strong certified unlearning guarantees without noisy training, and are broadly in line with those made in some other works on certified unlearning. To investigate the relevance of our methods beyond this setting, we conducted an *additional experiment* on a neural network with $2$ hidden layers of size $64$ and $32$, and ReLU activations, thereby moving to a *non-convex* and *non-smooth loss*. We report these experiments in [this anonymized repository](https://anonymous.4open.science/r/ICML2026_Submission16907-7E9D/rebuttal_NN_digits.pdf). We observe superior performance of $VRU$ across $r_f$ values both on privacy and utility metrics. We believe this efficient adaptation of the algorithm to the non-convex setting is due to the fact that our gradient estimator $\tilde{\nabla}_t$ remains unbiased as long as the unlearning algorithm is initialized at any minimum  of the loss function, even a local one, or more generally at any critical point.
> While this does not establish applicability to the large-scale deep neural networks, it does suggest that the method may remain relevant beyond the convex setting.
>
> **Supplementary dataset.** We evaluated our method on an additional dataset to further widen the scope of our experiments. We reproduced Figure 2's experimental protocol on the Wisconsin Breast Cancer dataset from scikit-learn, applying *VRU* alongside the same empirical baselines. We measured excess risk for utility and U-MIA accuracy for privacy after unlearning with a budget of $5$ *Fine-Tuning* epochs. We report our results at [this anonymized link](https://anonymous.4open.science/r/ICML2026_Submission16907-7E9D/rebuttal_LR_WBC.pdf). Consistent with Figure 2, *VRU* outperforms its empirical competitors, even with $r_f=0.1$ in this setting. We observe the same conclusions: privacy leakage is worst at the intermediate $r_f$ value for most methods, and *Fine-Tuning* fails to make the forget set private for the smallest $r_f$ value, as it lacks an active ascent mechanism.
>
> **Limited $r_f$ range of applicability.**
> The reviewer correctly observes that, for *larger* values of $r_f$, some empirical baselines achieve a smaller excess risk than *VRU* in Figure~2. However, these are not the values of $r_f$ that are most relevant in practice: machine unlearning is primarily motivated by *small* forget fractions, typically corresponding to the removal of a single user's data or a limited number of training examples, for instance in response to individual deletion requests under regulations such as GDPR, i.e., $r_f \approx 1/n$. In precisely this practically central regime, *VRU* consistently outperforms all compared methods in both privacy and utility.
> For more moderate values of $r_f$, *VRU* still outperforms *NFT* in all tested scenarios, making it, to the best of our knowledge, the strongest certified unlearning method among those evaluated for any $r_f \leq 0.1$. More broadly, the fact that retraining-based approaches become preferable as $r_f$ grows is not specific to *VRU*, but reflects a structural aspect of certified unlearning. When $r_f$ becomes large, the retain dataset shrinks substantially, so retraining becomes increasingly efficient, while at the same time offering ideal privacy guarantees. In that regime, the computational advantage of certified unlearning over retraining can disappear altogether.

---

> > ### Author Rebuttal · Reviewer_fSLX · 2026-04-03
> >
> > Thanks for your answers. I am keeping my already above-average score.

---

### Official Review · Reviewer_8Fsg · 2026-03-17

**Soundness:** 4
**Presentation:** 3
**Significance:** 4
**Originality:** 3
**Overall Recommendation:** 5
**Confidence:** 4

**Summary:**

The paper introduces VRU (Variance-Reduced Unlearning), a first-order certified unlearning algorithm for strongly convex objectives. The core problem with existing (epsilon,delta)-unlearning methods is that they treat the forget set as a noise-calibration tool only, never as an optimization signal. VRU addresses this by building a variance-reduced gradient estimator inspired by SVRG: at each step, the standard stochastic retain gradient is corrected by a term anchored at the pre-trained optimum θ* involving a forget-set gradient sample. This correction is unbiased by the first-order optimality condition at θ*, and it reduces variance near θ* without requiring the periodic full-batch recomputation that makes SVRG expensive. The algorithm then runs projected SGD with a 1/(mu t) step size schedule and finally injects Gaussian noise to certify (epsilon, delta)-unlearning. Main claims: (i) VRU achieves O(1/e) dependence on excess risk, vs O(1/e²) for the prior best method NFT; (ii) this extends the regime where unlearning beats retraining; (iii) VRU asymptotically outperforms any first-order method that ignores the forget set in a specific error regime. Experiments on logistic regression corroborate the theory.

**Compliance With Llm Reviewing Policy:**

Affirmed.

**Final Justification:**

After the rebuttal and discussion I am happy to with the changes and clarification. Raising my score to 'accept'.

**Key Questions For Authors:**

1. Oracle complexity. Theorem 4.1 and Corollary 4.2 compare VRU and NFT in terms of iteration count, but VRU uses roughly 3-4x more gradient evaluations per iteration (gradients at θ_t and θ* for both a retain and a forget sample). Does VRU still improve upon NFT when the comparison is made in terms of total gradient oracle calls? If so, please state this explicitly; it would materially strengthen the theoretical contribution.
2. Sensitivity to inexact θ.* The algorithm requires the exact pre-trained optimum θ*, which is used both as the anchor for variance reduction and as the center of the projection ball. In practice, training stops at an approximate optimum with some residual error. How does the convergence guarantee and (ε,δ)-unlearning certificate degrade as a function of ‖θ_init - θ*‖? The authors note robustness to inexact initialization empirically, but there is no theoretical treatment of this case.
3. Scope of the separation theorem. Theorem 4.4 guarantees that VRU outperforms any forget-set-free method, but only in the regime e < ckappa²_eps,delta (rf/(1-rf))² e0. Is this a tight characterization of when the separation holds, or just a sufficient condition from the proof technique? For typical values of kappa_eps,delta and say rf = 10^{-2}, what does this regime look like relative to e_0? A brief discussion, or a numerical illustration, would help readers understand when to expect the benefit in practice.
4. Comparison to Chien et al. (2024) and Koloskova et al. (2025). The theoretical and experimental comparison is to NFT only. Both Chien et al. and Koloskova et al. inject noise at each gradient step during fine-tuning, a different strategy. Does VRU's O(1/e) rate improvement also hold relative to these methods? If not, the claim of beating the state of the art should be scoped accordingly.

**Limitations:**

The paper's limitations section mentions strong convexity and the requirement of knowing θ* exactly, but there is another practical limitation not discussed: the algorithm as stated requires knowledge of the strong convexity parameter mu (for the step size 1/(mu t)) and the Lipschitz constant L (for the projection radius, at least in the theoretical version of the algorithm). The practical version (Algorithm 2) replaces L with the forget-set gradient norm, which is a nice workaround, but mu is still needed. Do other certified unlearning methods face the same issue, and what happens if mu is misspecified? Could an incorrect projection radius compromise the convergence or the (epsilon,delta) guarantee? This deserves at least a brief discussion.

**Strengths And Weaknesses:**

This is a technically solid paper with a clean central idea. I am recommending acceptance. Some concerns follow. If my questions are answered, I'm happy to increase my score from weak accept to accept.

Soundness
The gradient estimator is unbiased by the optimality condition at θ* (Eq. 4-5), and its Lipschitz constant is bounded via Lemma A.2. The proof of Theorem 4.1 applies Rakhlin et al. (2011)'s result on the last iterate of projected SGD and then uses the Gaussian mechanism. Lemma A.5, showing how high-probability sensitivity bounds compose with DP, is a clean addition not easily found elsewhere.
A concern: the O(1/e) rate improvement is stated in terms of iteration count T, but each VRU iteration requires gradient evaluations at both the current iterate and the anchor θ*, for both a retain and a forget sample, so roughly 3-4x the per-iteration cost of NFT. The theoretical comparison in Theorem 4.1 and Corollary 4.2 does not account for this. Whether VRU still wins on total oracle complexity is not addressed, and it should be.

Theorem 4.4 (the separation result) is the paper's most striking claim but leans heavily on a borrowed lower bound from Van Waerebeke et al. (2025): the proof essentially combines VRU's upper bound with the existing lower bound for forget-set-free methods and takes a ratio. This is logically valid but somewhat indirect. More importantly, the result only holds in the regime e < cκ²_ε,δ (rf/(1-rf))² e₀. How restrictive this regime is in practice is not discussed; it would help to have a numerical illustration in the paper's own experimental setting.

Presentation
Generally well written and the two-phase structure of the algorithm (projected SGD then noise) is transparent. Section 4.5 honestly addresses the Mavrothalassitis et al. (2025) negative result on descent-ascent methods, which I appreciated.

One clarity issue that made reading noticeably harder: the quantity e₀ appears in every main theorem (Theorem 4.1, Corollaries 4.2, 4.3, Theorem 4.4) but is never explicitly defined in the paper. It is the initial excess risk L(θ*, Dr) - L(θ*r, Dr), inherited from Van Waerebeke et al. (2025), but a reader who has not read that paper closely will be lost. Please define e₀ explicitly, and ideally give an explicit bound in terms of rf and the loss parameters to make the results self-contained.

Additional presentation issues:
1. Line 252: "One highlight A key advantage" is a clear editing error.
2. Lines 255-256: the word "typically" appears twice in adjacent sentences.

The experimental section validates the theory in the strongly convex setting, which is appropriate. Experiments on a small neural network would be interesting, though I do not consider this a prerequisite for acceptance given the paper's clearly theoretical scope.

Significance
The gap being closed is real and practically meaningful: certified methods have consistently underperformed empirical ones in benchmarks largely because they ignore the forget set. Bridging that gap with formal guarantees is a genuine contribution. The O(1/e) vs O(1/e²) improvement is qualitatively important as it extends the regime where unlearning is provably cheaper than retraining. The restriction to strongly convex objectives limits the immediate practical reach, but the paper is upfront about this.

Originality
The key insight, exploiting the first-order optimality condition at θ* to construct an unbiased, low-variance correction using forget-set gradients, is elegant and to my knowledge novel in the certified unlearning context. The connection to SVRG is explicit and well-motivated, and the paper cleanly distinguishes VRU from SVRG: the periodic full-batch recomputation that makes SVRG expensive is replaced by a cheap stochastic forget-set term with the same expectation, a structural observation specific to the unlearning setting. The related work coverage of the certified unlearning landscape (Neel et al., Van Waerebeke et al., Koloskova et al., Chien et al.) is adequate.

---

> ### Author Rebuttal · Authors · 2026-03-30
>
> **Fair comparison across methods.** We thank reviewer 8Fsg for appreciating our theoretical contribution and the originality of composing high-probability convergence bounds with DP. As for our theoretical analysis, the provided bounds use the Landau notation, which does not account for constant multiplicative factors. Since VRU improves the dependence on $e$ from $O(1/e^2)$ to $O(1/e)$, the 3x per-iteration overhead does not affect the asymptotic comparison: for small enough $e$, *VRU* outperforms *NFT* in total gradient oracle calls as well. We also observe that  worst-case analysis generally introduces large and unrepresentative factors that have little practical relevance, such as the value $h(T,\delta)$ inherited from Prop. $1$ of Rakhlin's work. At the same time, constant factors may play an important role in practice and we have addressed this potential issue experimentally.
> All of our experiments are carried-out with an *identical number of gradient oracle calls* for each method, as stated in L318-323 (right), and discussed in further detail in our response to Review fSLX.
>
> **Scope of experiments.** We provide an additional experiment on a neural network in our response to Review fSLX.
>
> **Robustness to inexact initialization.** We thank the reviewer for their interesting theoretical question. Initializing unlearning algorithms at the minimum of the loss function is a common assumption, notably in the second-order certified unlearning literature (e.g. [1]). If that assumption does not hold and we initialize our algorithm away from the global optimum $\theta^*$, this has two main consequences, as stated by the reviewer: the projection ball is not centered on the correct point, and the gradient estimator is affected by an additional bias term. Regarding the projection ball, we believe this is not a significant issue, as its radius could be increased to account for potential inaccuracies. This would increase worst-case running time by a constant factor but would not affect the rest of the theoretical analysis. Figure 3 also suggests that removing the projection step entirely does not impact the algorithm's performance.
> However, the second part is more challenging, since the gradient estimator $\tilde{\nabla}_t$ would then become biased, which prevents us from applying Rakhlin's results. Still, it is possible that unlearning guarantees could be established in this case, by leveraging the recent work [2], which provides high-probability convergence results for strongly-convex functions under biased gradient estimates. Extending our analysis in this direction would require substantial additional work, but it appears to be a promising path towards handling  the non-exact initialization setting. We thank the reviewer for this remark and we will add this discussion to the paper.
>
> [1] "Approximate Data Deletion from Machine Learning Models", Z. Izzo et al., 2020
>
> [2] "Accelerated Gradient Methods with Biased Gradient Estimates: Risk Sensitivity, High-Probability Guarantees, and Large Deviation Bounds", M. Gürbüzbalaban et al., 2026.
>
> **Comparison to other SOTA methods.** The results from Chien et al. (2024) and Koloskova et al. (2025) address unlearning under weaker assumptions on the loss and analyze the number of iterations required to unlearn, but do not characterize the level of excess risk achieved after unlearning. Even if such a characterization were provided, both methods do not access the forget set at unlearning time and are therefore asymptotically less efficient than *VRU* in the low-error regime, as implied by Theorem 4.4. Our claim of outperforming the state of the art is thus justified. We will make this distinction clearer in the next revision of the manuscript.
>
> **Lack of definition for $e_0$.** We thank the reviewer for spotting this and for reading carefully enough to infer its meaning. We will define the bound on the initialization error $e_0 := L^2/\mu$ in Section 3, after Assumption 3.1, and correct the other two presentation issues as well.
>
> **Width of Theorem 4.4's error regime and reliance on previous results.** We discuss these elements in our response to Reviewer T71M.
>
> **Robustness to unknown constants.** We are happy to expand on this question and to include this discussion in our paper. Strictly speaking, *VRU* requires knowing $\mu$, but this limitation is shared by other certified methods (e.g., *NFT* needs both $L$ and $\mu$), and the use of these constants is ubiquitous in optimization papers. While empirical estimation techniques exist for $\mu$, their interplay with unlearning remains unexplored. In practice, *VRU*'s variance-reduction formulation tolerates much larger initial learning rates than empirical counterparts ($\eta_0 \approx 1$ vs. $10^{-3}$, Fig. 2), making it robust to $\mu$ misspecification. Moreover, Figure 3 shows that removing the projection step altogether does not impact performance, further reducing sensitivity to misestimated $\mu$ and $L$.

---

> > ### Author Rebuttal · Reviewer_8Fsg · 2026-04-03
> >
> > My concerns have been discussed sufficiently for me to raise my score to 'accept'.

---

### Decision · Program_Chairs · 2026-04-30

**Decision:**

Accept (regular)

**Comment:**

This submission studies the theoretical understanding of machine unlearning in a simplified setup with strongly convex smooth losses and various other assumptions (like disjoint retain and forget sets). The main focus is to highlight the advantage of using the forget set in the unlearning process. To the best of my understanding, the main theoretical improvements rely on two ideas:
- Replacing SGD for fine-tuning on the retain set with SVRG, which improves the convergence on convex functions from $O(1/\sqrt{T})$ for $T$ iterations of SGD to $O(1/T)$ for SVRG.
- Using the idea (assumption) that the gradient at the original optimum is exactly zero, and thus obtaining an unbiased variance reduced retain-set fine-tuning gradient by anchoring around the optimum. This retain-set gradient estimate involves the full-set gradient at the original optimum, which in turn brings in the gradient on the forget set, thereby "introducing" the forget set into the unlearning.

Leveraging the faster convergence of SVRG, the usual certifiable unlearning analyses are built upon to highlight a separation between the proposed scheme and existing certiable algorithms, with the main storyline being that the use of the forget set in the unlearning process (via this anchored variance-reduced stochastic gradient) allows for this separation. However, based on my own assessment, I think that a critical "theoretical ablation" is missing here -- it seems to me that applying SVRG to the retain-set fine-tuning with any anchor (as done in the original SVRG algorithm) should still provide the faster convergence guarantees than SGD, and thus the separation results. The gradient at this anchor can be completely independent of the forget-set gradient, and can be made unbiased as in the original SVRG algorithm without any use of the forget-set. The use of the original optimum as the anchor is a neat way of introducing the forget-set gradient in the unlearning, but it is not clear if it is necessary for the main theoretical advantages shown here (as many of the computational overhead in the use of a different anchor will not affect the rates in terms of asymptotics). This comment is solely based on my own assessment and not a weakness brought up by any of the reviewers.

Another issue (again based on my own assessment and not brought up by the reviewers) is that the use of the original model (which was trained on the forget set) in every single gradient update during the unlearning seems to practically introduce signficant opportunity for forget-set information leakage. The proposed scheme (and similar existing ones this is based on) rely on the Gaussian mechanism to guarantee certified unlearning, but the Gaussian mechanism is usually not practical beyond just linear models. It would seem to me that membership inference attacks on larger models unlearned by anchoring on the original model might have a better attack success rate.

Beyond the theory, the empirical evaluations are limited to convex losses / linear models, with some preliminary experiments on small neural networks in the rebuttal. While these experiments are informative, unlearning with strongly convex linear models is usually a very simple practical problem. The main certified unlearning guarantees is through the final Gaussian mechanism, which is usually fine with linear models, but usually significantly degrades the utility of a moderately sized neural network. The ease of unlearning on linear models is highlighted by the really favorable MIA accuracies that linger very close to 50% (with the worst being 55% in figure 2). Thus, I would think that the unlearning/retain-utility of larger models would be much more unfavorable than the linear models given the potential for information leakage in larger models and strong adverse effect of the Gaussian mechanism on large neural networks.

Overall, I think this submission, with the main contributions being theoretical, is missing a critical theoretical analysis to validate the main claim that introduction of the forget-set allows for a separation in guarantees. A more thorough empirical evaluation on larger models would further improve this submission. Thus, I would recommend a weak accept at best since the reviewers all unanimously recommended acceptance.